# Exposed soil and mineral map of the Australian continent revealing the land at its barest

Dale Roberts [1]*, John Wilford[2]* & Omar Ghattas [1]*

Multi-spectral remote sensing has already played an important role in mapping surface mineralogy. However, vegetation – even when relatively sparse – either covers the underlying substrate or modifies its spectral response, making it difficult to resolve diagnostic mineral spectral features. Here we take advantage of the petabyte-scale Landsat datasets covering the same areas for periods exceeding 30 years combined with a novel high-dimensional statistical technique to extract a noise-reduced, cloud-free, and robust estimate of the spectral response of the barest state (i.e. least vegetated) across the whole continent of Australia at 25 m$^2$ resolution. Importantly, our method preserves the spectral relationships between different wavelengths of the spectra. This means that our freely available continental-scale product can be combined with machine learning for enhanced geological mapping, mineral exploration, digital soil mapping, and establishing environmental baselines for understanding and responding to food security, climate change, environmental degradation, water scarcity, and threatened biodiversity.

[1] Australian National University, Acton, ACT 2601, Australia. [2] Geoscience Australia, Symonston, ACT 2609, Australia. *email: dale.roberts@anu.edu.au; john.wilford@ga.gov.au; omar.ghattas@anu.edu.au

Most natural terrestrial land surfaces consist of mixtures of vegetation, soil and bedrock. These materials reflect and absorb radiation across different wavelengths resulting in an observation of a spectral response. Our ability to accurately distinguish the surface materials from these spectral signatures depends on the spatial and spectral resolution of the instrument, the state of the atmosphere and homogeneity of the surface relative to the resolution of the sensor. However, as green and dry vegetation both strongly interact with electromagnetic radiation across the same wavelength region that is used for mineral mapping[1–3], the ability to separate – or at least recognise – the influence of the vegetation is critical to the success of mapping the characteristics of soil and rock. Indeed, just 10–30% vegetation coverage can significantly impede the recognition of mineral spectral features[1,2,4,5].

Where geo-botanical relationships are strong (i.e. plant species or communities are strongly linked to the type of bedrock or soil), indirect measures can allow inferences to be made on the nature of the substrate[6,7]. However, in the absence of geo-botanical inference, effective mapping of soils using remote sensing is restricted to areas where the vegetation cover is either sparse or absent due to drought or cultivation[8]. As such, most remotely sensed mineral mapping studies have covered areas of relatively sparse vegetation cover[9–12]. In Australia, a significant obstacle to geological remote sensing are fire scars in the arid zones as they create a complex pattern of vegetation densities that makes it difficult to map bedrock and soil[13].

Recently, some studies have attempted to exploit the full time series archive of Landsat observations, collected over the last 30 years, to produce per-pixel mosaics of the barest earth (i.e. exposed soil)[14,15]. However, their approaches were only attempted in small geographic areas and their methodologies are based on user-defined thresholds and data-mining techniques that are unlikely to be sufficiently complex and scalable to correctly remove all the non-bare responses in the observations. This is especially true for a large continent such as Australia, which exhibits a very diverse range of climates, terrestrial land surfaces

and biophysical changes through time. Scaling methods to a continental archive of data presents a major challenge in the analysis of big datasets called heterogeneity whereby outliers and various states are no longer sparse but become proper sub-populations in the data that are difficult to disentangle[16].

In this paper, we tackle these technical issues and provide the first continental-scale mosaic of Australia at its barest state using the full temporal archive of Landsat observations. We achieve this by proposing a statistical estimator of the barest spectra that is both robust to contamination (such as cloud cover, shadows, detector saturation and pixel corruption) and, most importantly, correctly maintains the relationship between all the spectral wavelengths enabling the application of machine learning and spatial statistics to further separate vegetation and mineral spectra[17,18]. The result of our approach is the generation of two continental-scale composite Barest Earth mosaics of Australia, one of shorter temporal depth using only Landsat 8 observations and the other using the full 30 + year archive combining Landsat 5, 7 and 8.

**Table 1 Description and wavelength ranges (in micrometres) of the spectral bands of the sensors aboard the three Landsat satellites used in this study. The Landsat-8 OLI sensor has a higher radiometric resolution of 12-bit resulting in higher dynamic range and a greater ability to characterise land cover state and condition.**

|  | Landsat-5 TM | Landsat-7 ETM+ | Landsat-8 OLI |
|---|---|---|---|
| BLUE | 0.45–0.52 (B1) | 0.45–0.52 (B1) | 0.45–0.51 (B2) |
| GREEN | 0.52–0.60 (B2) | 0.52–0.60 (B2) | 0.53–0.59 (B3) |
| RED | 0.63–0.69 (B3) | 0.52–0.60 (B3) | 0.64–0.67 (B4) |
| NIR | 0.76–0.90 (B4) | 0.77–0.90 (B4) | 0.85–0.88 (B5) |
| SWIR1 | 1.55–1.75 (B5) | 1.55–1.75 (B5) | 1.57–1.65 (B6) |
| SWIR2 | 2.08–2.35 (B7) | 2.09–2.35 (B7) | 2.11–2.29 (B7) |
| **Radiometric resolution** | 8-bit | 8-bit | 12-bit |

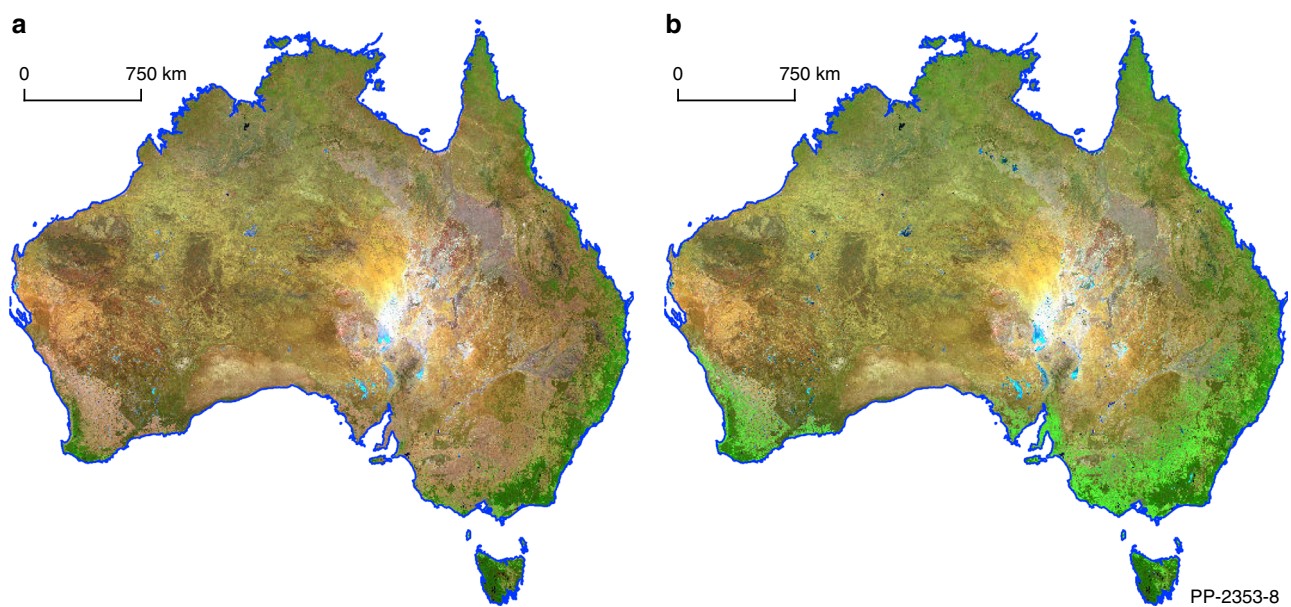

**Fig. 1** Comparison of the Landsat-8 Barest Earth mosaic to its inverse, most vegetated state, to highlight the reduction in vegetation influence. The mosaics are coloured using shortwave infrared 1.57–1.65 μm, near infrared 0.85–0.88 μm, green 0.53–0.59 μm in the red, green and blue image channels. **a** Barest Earth Landsat-8 mosaic, **b** Most Vegetated Landsat-8 mosaic generated by inverting the Bare Earth algorithm to get the opposite response. Remaining green areas in the Barest Earth mosaic indicate areas of persistent vegetation.

## Results

**Data and approach.** The approach starts by considering ~16 billion 6-dimensional time series across the continent of Australia where each time series represents the variation of the BLUE, GREEN, RED, NIR, SWIR1 and SWIR2 wavelength bands in a 250-m$^2$ pixel (see Table 1). These time series were generated from imagery of the Australian continent collected by the Landsat series of satellites over a period of more than 30 years. These images were geographically aligned and stacked, so that the pixels line up through time, to obtain a spatial-spectral-temporal (tensor) data set. We processed all data using the Open Data Cube (previously known as Australian Geoscience Data Cube[19,20]). Differences between Landsat sensors were minimised by ensuring that all observations were atmospherically, BRDF and topographically corrected to measurements of surface reflectance using the NBAR/T approach[21]. This surface reflectance correction, that uses several ancillary datasets to get a highly accurate atmospheric reading, is only available over the Australian continent.

For each time series, our aim is to obtain an estimate of the barest state observed relating to either soil or exposed rock. To achieve this, we holistically consider each pixel observation through time as 6-dimensional vector. The first ingredient of our method is to apply a high-dimensional statistic called a weighted geometric median (WGM) that combines ideas from the classic (one-dimensional) weighted median[22] with the high-dimensional geometric median (GM) proposed by Weber[23]; see methods. Recently, the GM has attracted significant interest in the mathematical community due to its remarkable robustness and confidence boosting properties[24–27]. These properties and mathematical results easily extend to the weighted case (that we propose) as well. Interestingly, there are many ways to extend the concept of a median to higher dimensions[28] but the GM is one of the more mathematically tractable of those proposed extensions. Our motivation to include weights into the definition of the GM, giving the WGM, is to focus the WGM on observations through time that exhibit the least amount of vegetation and the high level of robustness of the WGM removes the influence of the other sub-populations in the data. A key point is that by weighting (as opposed to filtering) we do not throw away any information. This allows the resulting pixel composite mosaic to be completely cloud-free and for the spectral signature of vegetation to be minimised. This approach also behaves well where the vegetation never achieves bareness, such as in areas with dense tree cover, as we get a result for every pixel. In other words, if a pixel consistently remains green through time then the resulting mosaic will have a green response at that location. This is a useful characteristic if a spatial model is to be applied to the product as our mosaic does not containing any missing data. The second ingredient to our approach is the design of an optimal weighting scheme that ranks, as opposed to filtering, the bareness of pixels and separate out the bare states from others, if they are present; see methods. The weighting scheme was derived by fitting an ensemble of regression models based on spectral features over various training areas exhibiting a high variability of biophysical states. A single model was derived from this ensemble using a loss function that minimises vegetation response and variability over these areas. The WGM then robustly summarises this bare state for each time series as a 6-dimensional vector (pixel) and combining all these pixels gives the continental-scale mosaic (Fig. 1).

**Validation.** We validated our Barest Earth mosaic three ways. First, we considered a small test area in Canberra where grass is commercially grown using pivot irrigation and then partially harvested throughout the year. This gives observations through time ranging from partially exposed bare soil to partial grassy cover but where a completely bare crop circle is never observed

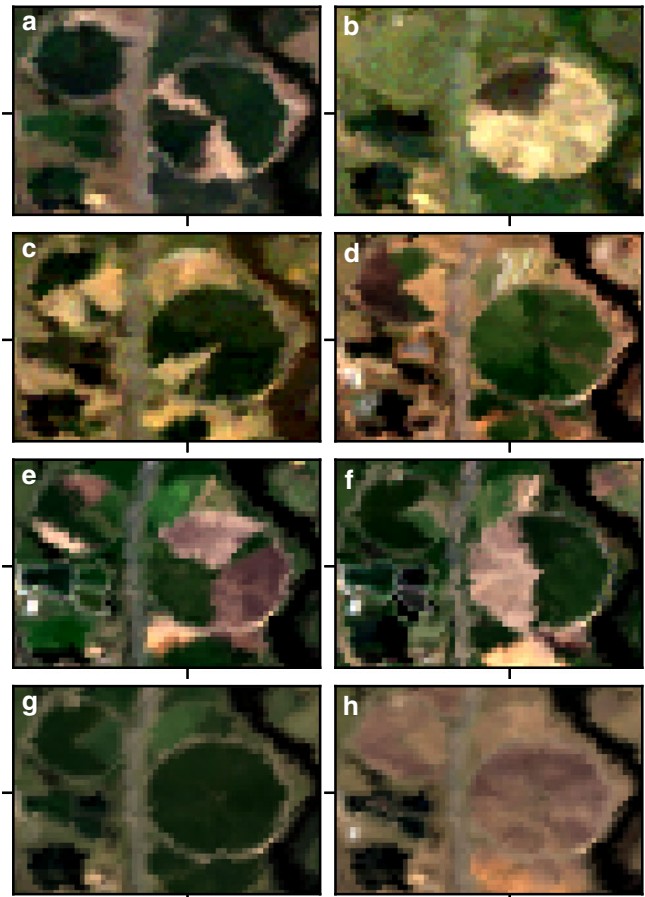

**Fig. 2** Test area at different time epochs compared to the Barest Earth and Most Vegated outputs. The local study area at 6 different observation times (**a**–**f**) compared to **g** the Most Vegated output and **h** the Barest Earth output (Landsat 30+ year). The Most Vegated output is generated by inverting the Barest Earth algorithm to get the opposite response. The observation times are: **a** 2006-02-10, **b** 2009-05-09, **c** 2011-08-26, **d** 2013-05-20, **e** 2016-02-06, **f** 2016-11-04. Images are displayed with red 0.64–0.67 μm, green 0.53–0.59 μm and blue 0.45–0.51 μm in the red, green and blue image channels. This area is located at 35°18′49.0″S 149°10′23.6″E. We used a 12-year stack (January 2005–January 2017) comprised of 616 scenes.

for any given time period (Fig. 2). Focusing on specific pixels within this area, we show in Fig. 3 the variation in spectral responses and our WGM estimate of the barest state. This gave confidence that our approach was able to separate the complex spatial-temporal-spectral dynamics into its constituents at each pixel to obtain a consistently bare crop circle. Most surprisingly, the resulting (synthetic) Barest Earth output in our test area is visually indistinguishable from a true observation and displays remarkable spatial smoothness even though the approach does not use any spatial information. This is consistent with recent findings[29].

The second part of our validation consisted of generating two continental-scale composite Barest Earth mosaics of Australia. The first, Landsat-8 Barest Earth mosaic, only uses Landsat-8 OLI (Operational Land Imager) observations (from 2013 to the present day), and the second, Landsat 30 + Barest Earth, uses a deeper time series from 16 March 1983 to the present day, with observations that combine Landsat-5 Thematic Mapper (TM), Landsat-7 Enhanced Thematic Mapper + (ETM+), and Landsat-8 OLI. Our hypothesis was that the mosaic generated from the longer

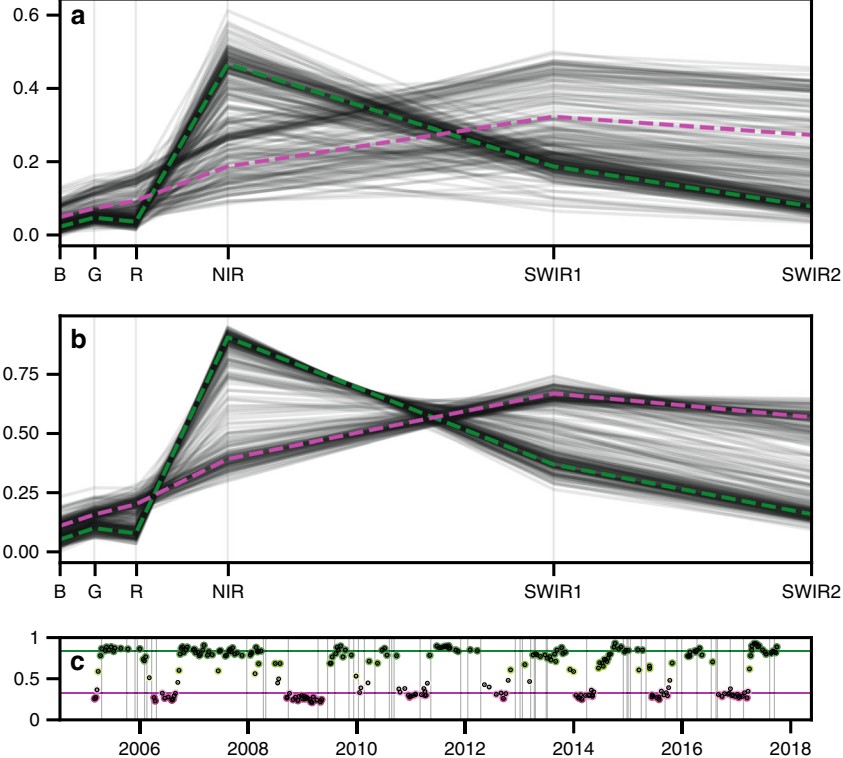

**Fig. 3** Spectral responses through time at a specific pixel location compared to the estimated Barest Earth spectra. The pixel location is identified with tick markers in Fig. 3. The first plot shows the observation spectra through time with the Barest Earth (magenta) and Most Vegated (green) shown with dashed lines, the second plot shows the observation spectra but scaled (as $x/||x||$ for observation vector $x$ to remove parallel albedo shifts due to atmospheric effects). The Most Vegated output is generated by inverting the Bare Earth algorithm to get the opposite response. The third plot shows the normalised difference vegetation index (NDVI) time series of the observations coloured using the Barest Earth weighting model. Horizontal lines show the normalised difference vegetation index of the Barest Earth (magenta) and the Most Vegated (green). Vertical lines show the presence of missing data due to cloud masking or LS7 SLC-off gaps.

time series would have less vegetation response and less influence from burn scars as the algorithm had more opportunity to potentially observe a barer state over this longer time period. This hypothesis was validated by comparing the two mosaics in various areas across the continent. It is quite clear that the increased temporal depth allows us to overcome the difficulty in mapping fire prone landscapes that preserve vegetation patterns which correlate with fire scars of different ages and intensities[13]. We see that this effect is significantly reduced in the Landsat 30 + mosaic improving our ability to map soil and rock characteristics (Fig. 4).

The third stage of our validation exercise involved a comparison with existing datasets at Geoscience Australia to assess the effectiveness of the approach in minimising the influence of vegetation and to ensure the spectra are consistent with known geology, spectral endmembers collected in the field, and national vegetation maps. As part of the exercise, we used the National Geochemical Survey of Australia (NGSA) dataset containing spectra from surface soil samples[30]. These samples were homogenised from the upper 10 cm of soil and dried prior to being analysed in the laboratory using spectroscopy and then resampled to Landsat sensor band specifications. We compared the spectra from these soil samples ($n = 1059$) to the 25 m² resolution spectra obtained from the Barest Earth mosaic at each location. We show the improvement in spectral similarity between the Barest Earth spectra and the NGSA soil sample spectra across Australia compared with a clear Landsat observation exhibiting a high vegetation signal (Fig. 5). We see little improvement in the arid area of Australia (<450 mm of average annual rainfall) due to the sparse amount of vegetation

present. However, in the non-arid area which has a high density of permanent and seasonal vegetation, we see significant improvement in matching the soil spectra (Fig. 6).

## Discussion

Our Barest Earth products show a significant reduction of the influence of vegetation in the multi-spectral bands with the result of enhancing our ability to observe responses pertaining to variations in surface mineralogy. The reduced influence in vegetation is most striking in landscapes with seasonal variations in ground cover (Fig. 7) and in more arid fire prone landscapes (Fig. 4). This allows us to overcome in many landscapes the largest challenges (vegetation masking and burn scars) in mapping soil and rock from space.

At one extreme, we have dense vegetation cover, corresponding to closed forest and closed shrub lands. In these areas, it is unlikely that exposed ground will be seen unless the vegetation is removed by land clearing. In areas of less dense vegetation cover (open woodlands where the crowns of trees do not touch, open shrub lands and grasslands), we have a much higher probability of estimating the bare spectral response through the time series. Periods of increased bareness commonly relate to seasonal drying, drought, the phenological cycle, fires, erosion and land clearing. For example, in agricultural landscapes where there is a pressing requirement to map and monitor soil health[31] we see a dramatic increase in bareness an consequently our ability to map soils directly (Fig. 7). A less obvious, but apparent at the 25 m² resolution, is a reduction in vegetation response in areas of relatively

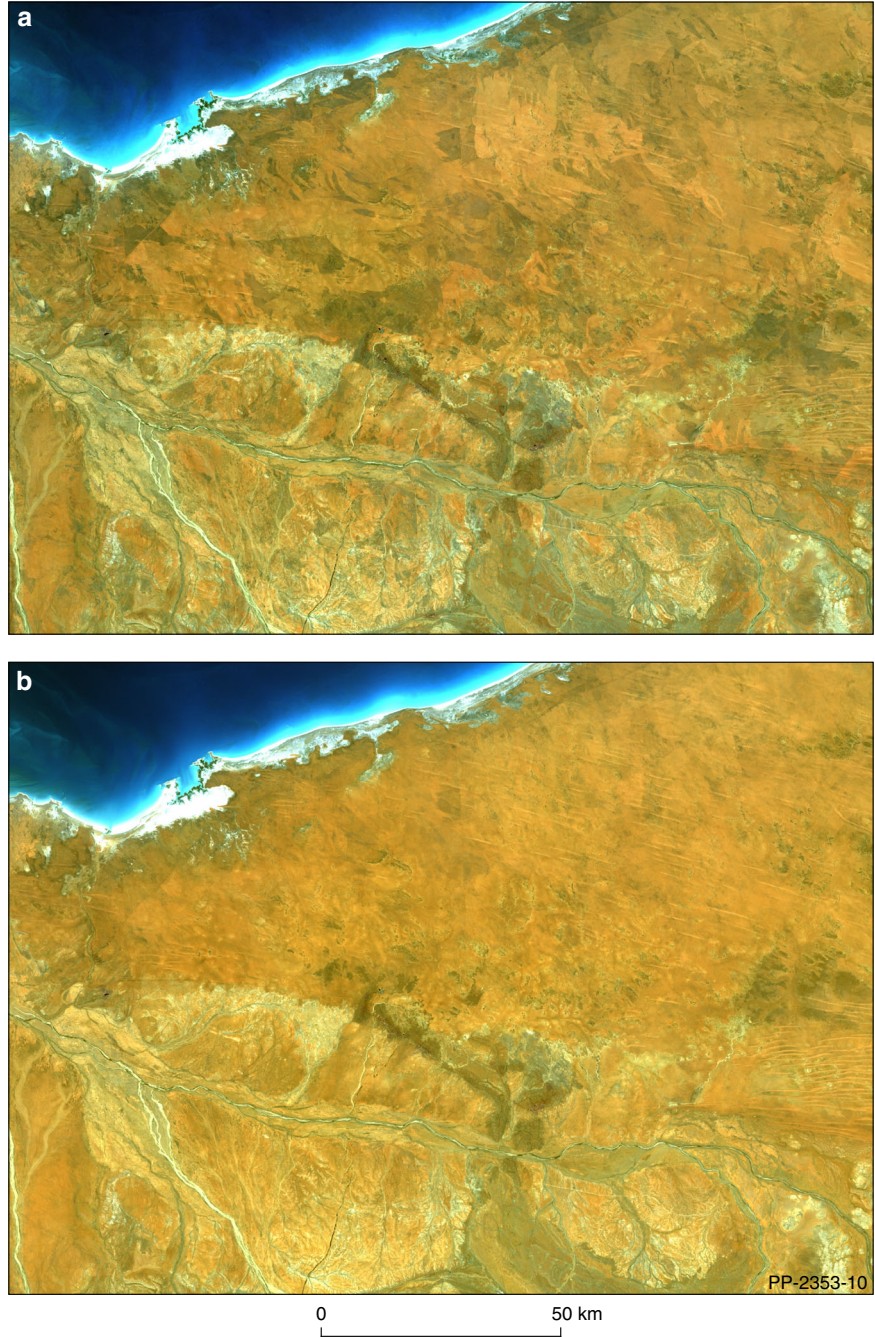

**Fig. 4** Comparison between the Landsat 8 Barest Earth and the Landsat 30+ Year Barest Earth mosaic over the north-west of Australia. The second phase of our validation consisted of a continental-scale comparison between the two products that have different temporal ranges for the input data. We show both images in true colour (red 0.64–0.67 μm, green 0.53–0.59 μm, blue 0.45–0.51 μm in the red, green, blue image channels). **a** Landsat 8 Barest Earth product exhibiting the vegetation patterns in the landscape (dark and light) that correlate with fire scars of different ages and intensities. **b** Landsat 30 +Year Barest Earth product showing the removal of the effect of fire scars.

low vegetation density corresponding to the central arid and semi-arid zone (Fig. 6). These arid zones cover ~70% of the Australian continent and, on average, have vegetation densities of the order of 10–30%, with local areas up to 60% being common[13].

From a soil and geological perspective, especially in areas where rocks and soil are exposed, multi-spectral remote sensing with Earth observation satellites has proven effective in mapping lithologies (e.g. individual or groups of minerals), geochemistry, bedrock structure and the nature of the regolith (i.e. weathered bedrock and sediments)[9,17,32–34]. Subtle geochemical alteration

patterns associated with mineral deposits have been also mapped using satellite imagery[9,35–37]. Mineral spectra associated with soils have been used to identify different soil types[38] and provide proxies to infer soil properties, such as cation exchange capacity[39]. However, these remote sensing approaches have always been limited by vegetation cover and burn scars in the landscape. Our continental-scale Barest Earth products now provide the ability to more accurately map soil and rock spectra over large areas that were previously masked by vegetation. Importantly, due to the multivariate nature of our technique, the spectral integrity

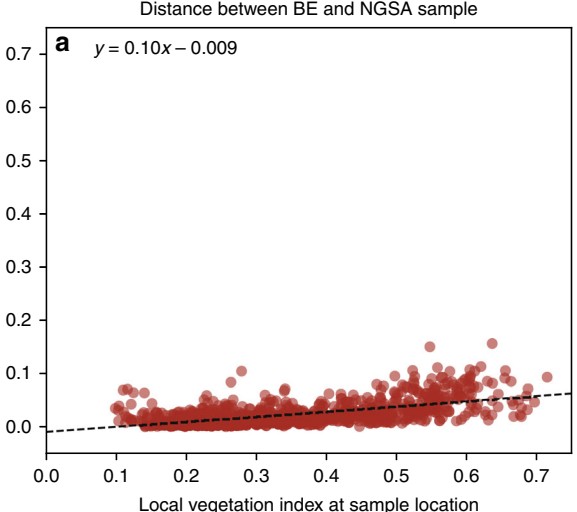

Distance between BE and NGSA sample

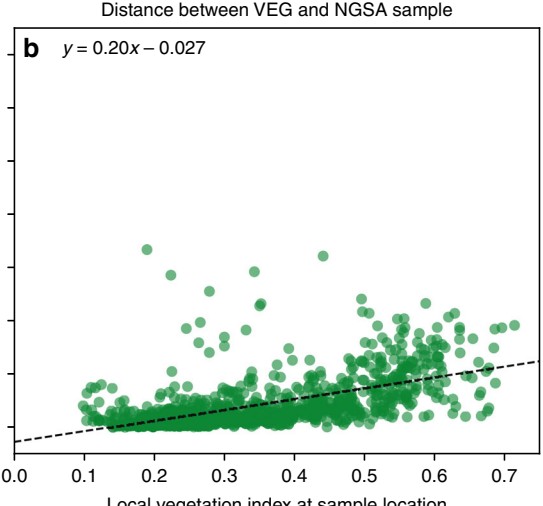

Distance between VEG and NGSA sample

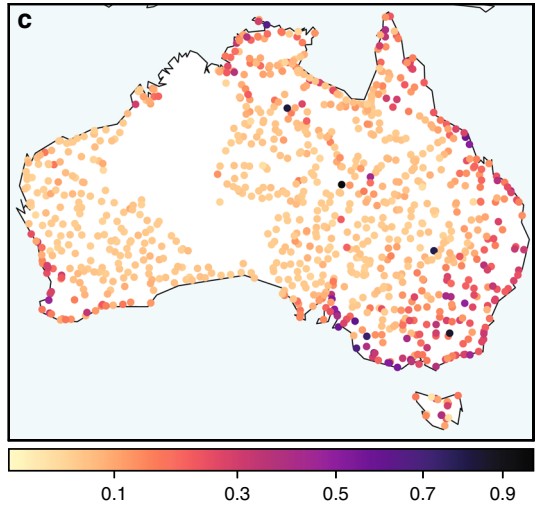

**Fig. 5** Distance between all the National Geochemical Survey Australia soil spectras to the Barest Earth and vegated spectras at the same locations. The cosine distance between the Barest Earth spectra and the NGSA soil sample (**a**) and between a clear observation exhibiting a high vegetation signal, denoted VEG, and the NGSA soil sample (**b**). We show this distance as a scatter plot with respect to a local vegetation index chosen as the maximum NDVI in a 75-$m^2$ neighbourhood (3 × 3 pixel) around the location. This shows, as one would expect, that areas with low vegetation have a small distance between the spectra of the Barest Earth, the clear observation and the NGSA soil sample. And vice versa, in areas with high amount of local vegetation we have a higher chance of obtaining a smaller distance between the Barest Earth spectra and the NGSA sample compared to the clear observation. This due to the fact that our Barest Earth model can never obtain a bare soil estimate in areas of permanent vegetation, for example. On the right, we show a map displaying the sample locations coloured based on the the difference between the cosine distance of the Barest Earth spectra to the NGSA spectra and the cosine distance of the VEG spectra compared to the NGSA spectra. We see that the areas showing the largest improvement are in the vegetated areas of Australia located outside the arid area center. We note that the NGSA dataset does not contain samples from some areas of Australia due to land access restrictions.

landscape. An advantage of our approach is that methods previously only effective on individual clear scenes with sparse vegetation cover can now be holistically calibrated and applied at a continental scale. This includes edge detection, band ratioing, and several statistical and spectral analysis techniques that are well-known for further separating and classifying vegetation and mineral spectra (e.g., directed principle component analysis[18], spectral unmixing[41], decorrelation stretching and saturation enhancement[17]). For example, see Fig. 8.

Our approach has produced the first Barest Earth mosaics for the Australian continent. They have broad application in mineral exploration, agriculture and in understanding the nature and processes operating within the life-sustaining critical zone between tree tops and groundwater aquifers[42]. These thematic maps will be powerful covariates in machine-learning algorithms used for predictive mapping of soil properties[43] and surface geochemistry[44] and will improve the detection of often subtle mineral alteration patterns associated with mineral deposits. These enhanced products will be integrated with other geological datasets that capture information from the deeper crust and upper mantle, as part of a national and holistic research initiative called UNCOVER[45] to support mineral exploration in Australia. Furthermore, as our method can be applied over any epoch of time, for example annually, it may be used to monitor bareness through time or deviations from its long-term observations can be assessed with direct applications in establishing environmental baselines, assessing impacts of climate change, and monitoring soil health and land degradation.

## Methods

**Weighted geometric median**. We consider our space-time stack of observations over the Australian continent as a collection of time series, where $\mathbb{X}_{ij}$ is the time series of observations at pixel location $(i, j)$ and $n := n_{ij}$ is the number of observations at that location. This is possible because all pixels are aligned through time due to the ortho-rectification and registration of all images in our archive. The observations in the $(i, j)$'th $p$-dimensional time series are written

$$\mathbb{X}_{ij} = \left[ \boldsymbol{x}_{ij}^{(1)}, \boldsymbol{x}_{ij}^{(2)}, \ldots, \boldsymbol{x}_{ij}^{(n)} \right]^T.$$

This means that $\mathbb{X}_{ij}$ can be viewed as a data matrix with dimension $n \times p$, wherein $p$ is the number of bands (so in the case of Landsat data, $p = 6$).

As our approach does not consider neighbouring values or the spatial context around each pixel, we may describe the model for a single time series without loss of generality, with the understanding that an identical process is carried out for

(e.g. relationships between bands) is maintained across all wavelengths. This makes possible the extraction of mineral features based on their spectral signatures using well-established band ratios and principal component analysis[11,17,40], or to apply machine learning algorithms for automated classification of mineralogy in the observed

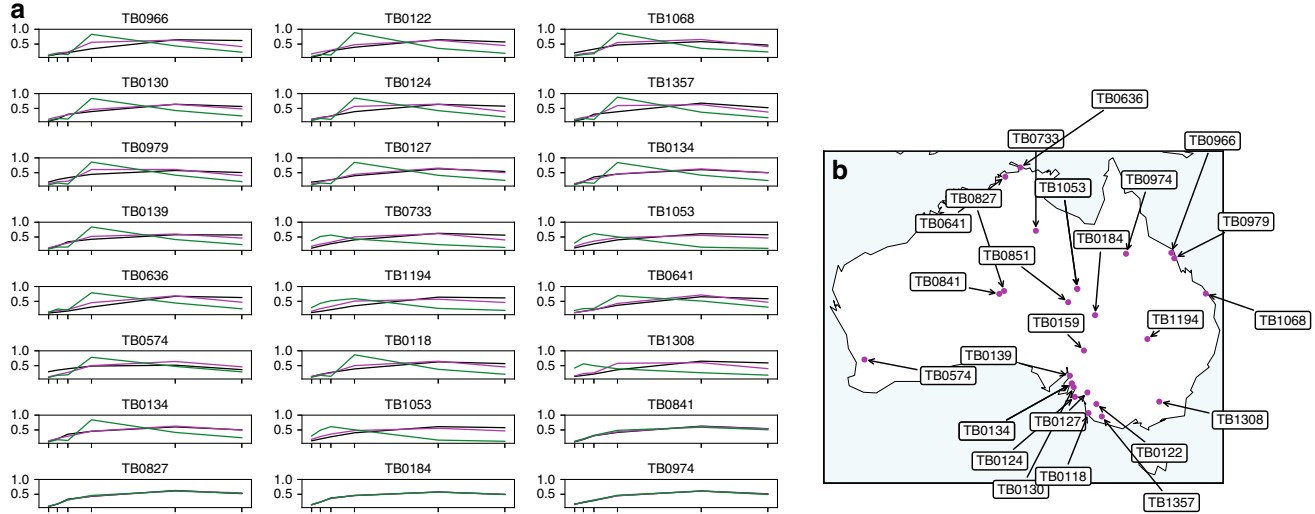

**Fig. 6** Comparison of soil spectra to the Barest Earth spectra and a vegetated observation at various locations across Australia. **a** A selection of the NGSA sample spectra (black) across the continent compared to the Barest Earth (magenta) and the clear vegetated observation (green). Spectra have been normalised as $x/\|x\|$ for observation vector $x$ to remove parallel albedo shifts. **b** Sample locations where the sample identifiers are given and can be referenced back to the NGSA dataset.

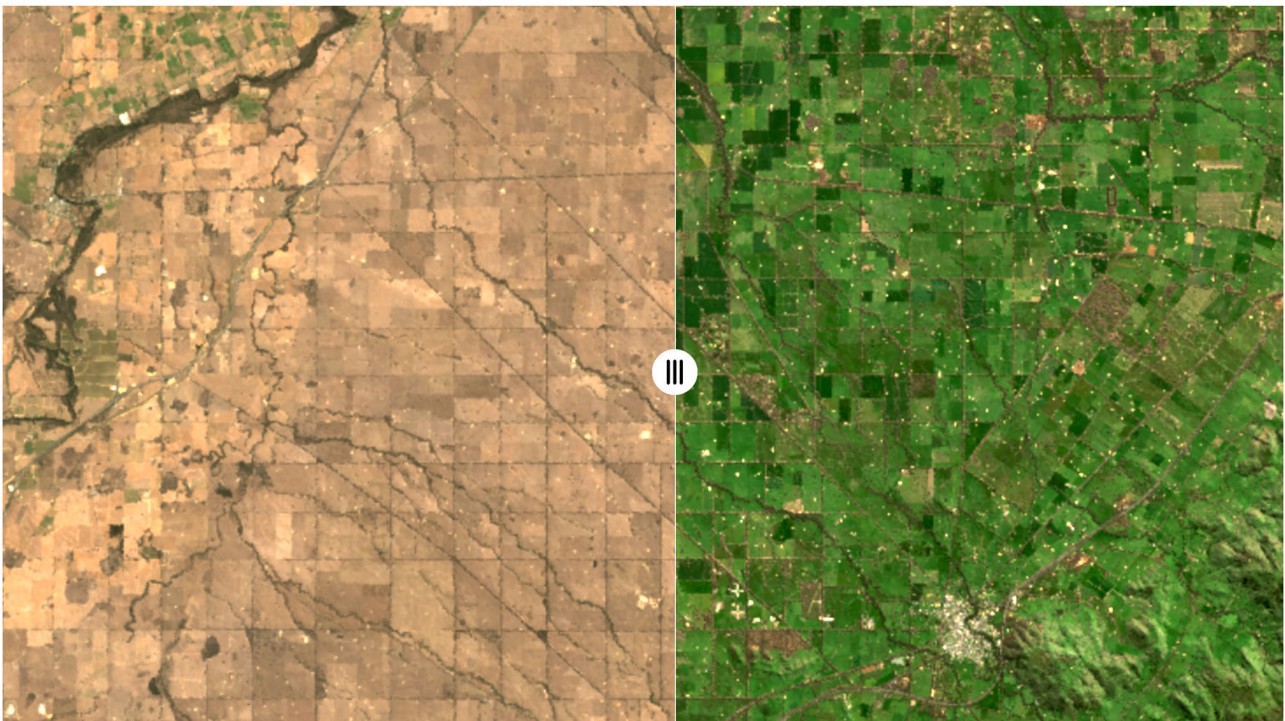

**Fig. 7** Comparison between the Barest Earth output and a single vegetated observation over an agricultural district in south-eastern Australia. Split screen view from National Map (http://nationalmap.gov.au/) showing the difference between our Barest Earth product (left) and (right) a single clear Landsat-8 observation (31/08/2019) in true colour. Location is 46.53550°S 145.18917°E.

each time series in our dataset. As such, we omit the "*ij*" subscript and write $\mathbb{X}$ for the time series (or data matrix) and $x^{(t)}$ for the $t$-th pixel observation in $\mathbb{X}$ to avoid notational clutter. Given $\mathbb{X}$, we compute a weighted geometric median (WGM) of the component observations by solving the following optimisation problem

$$\mathrm{m}_w := \operatorname{argmin}_{x \in \mathbb{R}^p} \sum_{t=1}^{n} w_t \| x - x^{(t)} \|$$

where $\|\cdot\|$ is the Euclidean norm on $\mathbb{R}^p$ and $w_t \in \mathbb{R}^p$ is the weight of the $t$-th observation. This is a generalisation of the geometric median (GM), which is obtained from the WGM when $w_t = 1$ for every $t$. The resulting quantity $\mathrm{m}_w$ is a

$p$-dimensional vector and, by following the mathematical proofs in Kemperman[46], it can be shown that $\mathrm{m}_w$ always exists for a set of observations. The weight function $w$ that assigns the weight $w_t$ for observation $t$ can be thought of as an unsupervised classifier that allocates a higher value to the biophysical state that we are interested in and a lower value to other states.

**Designing the weighting scheme.** The choice of weight $w_t \in [0, 1]$ allows us to penalise or accentuate certain observations through time at every pixel location $(i, j)$. Since the weight function can be viewed as an unsupervised classifier of the biophysical state, this implies that even though our overall methodology can be applied to any sensor, the weight function will need to be redesigned for each case.

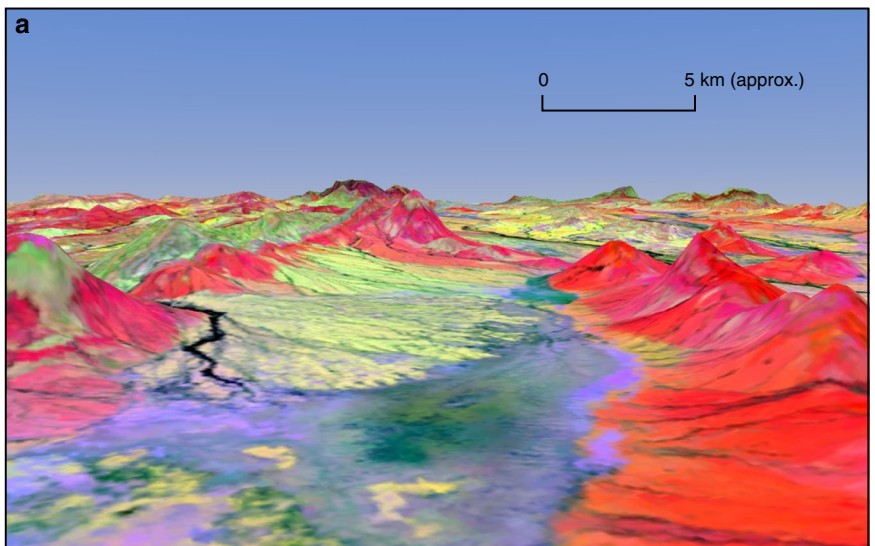

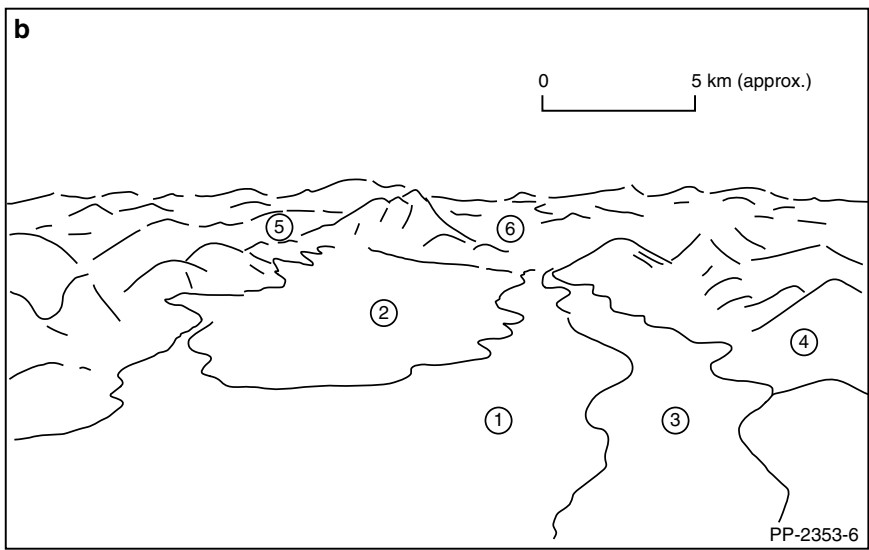

**Fig. 8** Perspective view of a enhanced Barest Earth output showing variations in surface mineralogy. We used enhanced mineral mapping techniques to produce a thematic mineral map of the Australian continent using the Barest Earth mosaic, a subset of north central Western Australia is shown above (**a**). The second principal component of ratio bands NIR/RED and SWIR1/SWIR2 is used to highlight clays (red); the SWIR1/NIR ratio represents iron oxides (green) and the addition of BLUE and SWIR2 represents highly reflective materials, such as silica-rich bedrock or quartz sand. This is perspective draped over a digital elevation model of the region. Key features (**b**) include: (1) alluvial sands and clays, (2) colluvial sediments including iron gravels, sand and clay, (3) colluvial clays, (4) proterozoic mudstone and siltstone, (5) proterozoic dolerite and gabbro and (6) reworked lateritic colluvial gravels. Mineral enhanced 3-D perspective drape with clays in red; iron oxides in green and highly reflective materials like silica-rich bedrock or quartz sand in blue.

In this paper, we consider the case of Landsat data; thus, we take $p = 6$ and so for a fixed spatial location $(i, j)$, the $t$-th observation, $x^{(t)} = \mathbb{X}$, in $\mathbb{X}$, is a vector of surface reflectance values relating to each sensor, i.e. $\mathbb{X} = [x_{\text{BLUE}} x_{\text{GREEN}}, x_{\text{RED}}, x_{\text{NIR}}, x_{\text{SWIR1}}, x_{\text{SWIR2}}]^T$. Our aim is then to identify a suitable, data dependent, weighting scheme that maps $\mathbb{X}$ to a value between $[0, 1]$. The weighting scheme consists of two steps: firstly the choice of an (unnormalised) weighting function $\tilde{w} - \mathbb{R}^p \rightarrow \mathbb{R}$, which is applied to each of the $n$ observations, and secondly a step that takes these transformed values $\tilde{w}(x^{(1)}), \ldots, \tilde{w}(x^{(n)})$ and normalises them relative to each other, so that the final weights through time $w_1, \ldots, w_n$ take values between 0 and 1, giving a weight vector $\mathbb{W} = [w_1, \ldots, w_n]^T$, where $w_t$ is the weight assigned to the $t$-th observation in the definition of the WGM. We note that a simple way to normalise values is to identify the minimum $w_{\min}$, and maximum $w_{\max}$ elements of $\tilde{w}(x^{(1)}), \ldots, \tilde{w}(x^{(n)})$ and to calculate $w_t = (w_t - w_{\min})/(w_{\max} - w_{\min})$ to ensure that weights are between 0 and 1; however, this does not lead to the most optimal result.

The first step is the choice of a weighting function $\tilde{w}$ that penalises or accentuates each observation based on its biophysical class at that point in time. We started by generating simple features with which to separate various biophysical classes, based on the spectral information in the pixel

$\mathbb{X} = [x_1, x_2, \ldots, x_p]^T$. In the most general case, these features have the functional form

$$f(\mathbb{X}) = \left( \frac{x_i - \mu x_j + \zeta}{\nu x_i + x_j + \xi} \right) \left( \frac{\alpha_1(x_1 - \gamma_1) + \cdots + \alpha_p(x_p - \gamma_p) + \rho}{\beta_1(x_1 - \sigma_1) + \cdots + \beta_p(x_p - \sigma_p) + \delta} \right) \quad (1)$$

for some choice of $i, j$ with $i \neq j$ and for vectors $\alpha = (\alpha_1, \ldots, \alpha_p)^T$, $\gamma = (\gamma_1, \ldots, \gamma_p)^T$, $\beta = (\beta_1, \ldots, \beta_p)^T$, $\sigma = (\sigma_1, \ldots, \sigma_p)^T$ and constants $\rho, \delta, \nu, \zeta, \xi, \mu$. Features of this form are often called ratio or normalised difference transformations of the bands and are commonly used in remote sensing. Classic examples include the Normalised Difference Vegetation Index NDVI $= (x_{\text{NIR}} - x_{\text{RED}})/(x_{\text{NIR}} + x_{\text{RED}})$, the Green Normalised Difference Vegetation Index GDVI $= (x_{\text{NIR}} - x_{\text{GREEN}})/(x_{\text{NIR}} + x_{\text{GREEN}})$, the Green Soil Adjusted Vegetation Index GSAVI $= ((x_{\text{NIR}} - x_{\text{GREEN}}))/(x_{\text{NIR}} + x_{\text{GREEN}} + 0.5) (1 + 0.5)$, the Ratio Vegetation Index RVI $= x_{\text{NIR}}/x_{\text{RED}}$ and the Soil Adjusted Vegetation Index SAVI $= (x_{\text{NIR}} - x_{\text{RED}})/(x_{\text{NIR}} + x_{\text{RED}} + 0.5)(1 + 0.5)$. We also considered more recent contributions to the literature on separating soil classes. The Ratio Index for Bright Soil (RIBS)[47] is a composite band ratio designed to

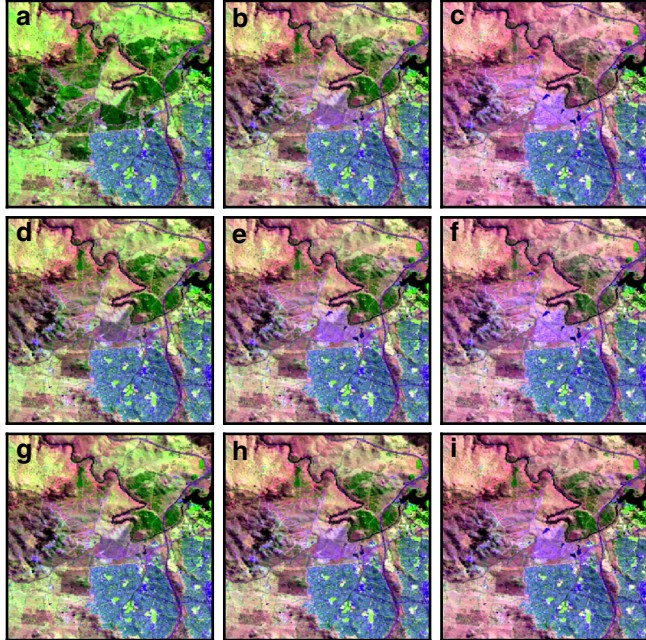

**Fig. 9** Comparison of the effect of different weighting schemes on the output of the proposed algorithm. The effect of making choices of weight function $\tilde{w}$ shown across an area to the west of Canberra, Australia. The location was chosen due to the presence of urban, water, vegetation and soil classes. The images are shown in false colour with SWIR1, NIR and BLUE in the RGB channels to accentuate healthy vegetation in bright green, soils in magenta and water in blue/black. The histogram stretches for all images are the same and chosen based on a 2–98% cutoff of the histogram of **b** which is an unweighted GM pixel composite mosaic. The weighting schemes are as follows: **a** is NDVI, **c** is $-1 \times$ NDVI, **d** is GNDVI, **e** is $-1 \times$ GNDVI, **f** is $-3 \times$ GNDVI, **g** is SAVI, **h** is $-1 \times$ SAVI and **i** is $-3 \times$ SAVI. We observe that **c**, **f** and **i** show the presence of the most soil across the image, however, **f** and **i** preference the presence of water (over soil) in some locations. NDVI provides good symmetry with respect to the GM case in **b**, in that **a** provides a highly vegetated composite and **c** a composite dominated by soil.

assign lower weights to bright soil land cover. It is given by RIBS = $x_{NDSI}/NTC1$ where[48] NDSI = $(x_{GREEN} - x_{SWIR1})/(x_{GREEN} + x_{SWIR1})$ and NTC1 is the normalised first Tasselled Cap Transformation[49]. The Product Index for Dark Soil[47] (PIDS) is designed to assign lower weight to darker soils and is given by PIDS = TCI × NDVI. We can then define our (unnormalised) weight function as

$$\tilde{w}(\mathbb{X}) = c_1 f_1(\mathbb{X}) + \cdots + c_m f_m(\mathbb{X}) \qquad (2)$$

where $c_1, \ldots, c_m$ are the constants to be determined and $f_1(\mathbb{X}), \ldots, f_m(\mathbb{X})$ are features (as defined above). The simplest cases are when $c_k = 1$ or $c_k = -1$ and $c_i = 0$ for $i \neq k$. Once the unnormalised weights $\tilde{w}(\mathbb{X}^{(1)}), \ldots, \tilde{w}(\mathbb{X}^{(n)})$ have been assigned to all the observations through time, we then normalise them using the softmax function, which gives

$$w_t = \frac{e^{\tilde{w}(\mathbb{X}^{(t)})}}{\sum_{i=1}^{n} e^{\tilde{w}(\mathbb{X}^{(i)})}}, \quad t \in \{1, 2, \ldots, n\}. \qquad (3)$$

This forces each $w_t$ to take values in the range [0, 1] and also results in all of the weights $w_1, \ldots, w_n$ summing to 1, so that (Eq. 2) is a convex combination of features. Figure 9 shows the effect of some model choices on the resulting pixel composite mosaic across an area to the west of Canberra (Australia).

**Fitting an optimal model**. Given $m$ features $f_1, \ldots, f_m$ of the form (Eq. 1), a model for bareness is defined in terms of our (unnormalised) weight function $\tilde{w}$ given by Eq. (2). This means that weights $c_1, \ldots, c_m$ need to be identified to obtain an optimal model. The difficulty is identifying a model that works well at a continental scale. Specifically, Australia has a large geographical size and extremely varied climate. The south-east and south-west corners have a temperate climate and moderately fertile soil. The northern part of the country has a tropical climate, varying between tropical rainforests, grasslands and desert. This means that a model that works well in some

areas may not be optimal in others. Moreover, due to the enormous range of soil types across Australia, developing a model that classifies bare soil can be difficult. Our model circumvents this ranking (instead of classifying) the observations through time at each pixel based on the relative presence of vegetation.

To reduce the potential of overfitting our weight model, we propose an approach that can be viewed as fitting an ensemble of regression models based on spectral features from which we derive a single model from this ensemble through a loss function that minimises average NDVI and variability over our training sites. We now describe this more precisely.

We fit an optimal model given features $f_1, \ldots, f_m$, in order to fine tune our approach to specific (smaller) regions or as a way to determine the optimal model for a continental-scale output. The fitting approach starts by identifying spatial areas that exhibit a large amount of spectral change through time. At the $i$'th area, obtain the spatial-spectral-temporal stack of observations $\mathbb{A}_i$. For simplicity, we explain our approach under the assumption that we only have one area $\mathbb{A}$ of dimensions $(n_y, n_x, p, n_t)$, where $p$ is the number of bands, $n_t$ is the number of observations through time and $(n_y, n_x)$ denote the number of spatial pixels.

Let $\theta = (c_1, \ldots, c_m)$ be the parameter vector for our choice of model $\tilde{w}$. We define our loss function, which we are trying to minimise, as

$$\ell(\theta; \mathbb{A}) = -\frac{\text{mean}_{y,x} \mathbb{B}}{\text{stddev}_{y,x} \mathbb{B}}$$

where $\mathbb{B} = -\text{NDVI}(\text{BARE}(\mathbb{A}; \theta))$. The function BARE, given a spatial-spectral-temporal stack $\mathbb{A}$, calculates the Barest Earth model using $\tilde{w}$, given parameters $\theta$ (as described in the previous section) and outputs a $(n_y, n_x, p)$-dimensional pixel composite mosaic (PCM). The function NDVI, given a $(n_y, n_x, p)$-dimensional image, calculates the Normalised Difference Vegetation Index NDVI = $(x_{NIR} - x_{RED})/(x_{NIR} + x_{RED})$ and returns a $(n_y, n_x)$ dimensional image. The functions $\text{mean}_{y,x}$ and $\text{stddev}_{y,x}$ calculate the mean and standard deviation across the $(y, x)$ dimensions and returns a number. As NDVI takes values in between $[-1, 1]$ and increases as vegetation increases, $\mathbb{B}$ will be an $(n_y, n_x)$ grayscale image with larger values at each pixel, meaning less vegetation (as we take the negative NDVI). This means that the $\text{mean}_{y,x}$ will increase as the PCM has reduced vegetation response. However, we also want to penalise outputs that appear more pixelated, so we divide by $\text{stddev}_{y,x}$. The term $\text{mean}_{y,x}\mathbb{B}/\text{stddev}_{y,x}\mathbb{B}$ is our reward or utility function, but we multiply this by $-1$ to obtain a loss function.

In the simple case where $m = 1$, we have $\theta = c_1$ and only one choice of feature $f_1$. Supplementary Fig. 1 shows the loss function $\ell$ for various choices of simple one-feature models. Remarkably, we observe a very smooth relationship between the parameter $\theta = c_1$ and our loss function $\ell$. This makes it very easy to identify the optimal $\theta$ that minimises the loss. The case of $m > 1$ is more complicated and involves minimising $\ell(\theta)$ over a $m$-dimensional surface $\theta = (c_1, \ldots, c_m)$. As we have (empirically) observed that $\ell$ is smooth in each coordinate of $\theta$, an efficient approach for minimising $\ell$ can be achieved through the use of Bayesian optimisation using a Gaussian process prior. Fitting over $N$ training areas simultaneously can be easily achieved by defining the loss function as $\ell(\mathbb{A}) = \ell(\theta, \mathbb{A}_1) + \cdots + \ell(\theta; \mathbb{A}_N)$ where $\mathbb{A}_1, \ldots, \mathbb{A}_N$ are the spatial-spectral-temporal stacks of observations for these $N$ training areas.

**Differences between the Barest Earth products**. Comparison of histograms between the two continental-scale products highlight some interesting differences. Landsat-8's OLI sensor provides improved signal-to-noise radiometric (SNR) performance quantised over a 12-bit dynamic range compared to the 8-bit dynamic range of Landsat-5 and Landsat-7 data. This results in 4096 potential grey levels in each band compared with only 256 in the 8-bit instruments on Landsat-5 and Landsat-7. This means there is trade-off. The improved SNR performance in Landsat-8 results in more accurate separation of vegetation and bare spectral signatures through time and the resulting Barest Earth mosaic maintains the improved SNR performance. Whereas, the Landsat 30+ product has worse SNR performance but a greater capacity to find barest ground due to the greater temporal depth.

**Model robustness**. Our approach for estimating the Barest Earth spectra is robust in a variety of ways. First, our features within our unnormalized weightings are robust to parallel shifts of the spectra due to the use of normalised difference ratios between the bands. This is important so that we are robust to atmospheric effects and residual thin cloud. Second, we recall that the breakdown point of an estimator is the proportion of incorrect observations (e.g. arbitrarily large observations) an estimator can handle before giving an incorrect (e.g., arbitrarily large) result. The geometric median is well-known to have a breakdown point[50] of 50%. We have conducted a simulation study of our model, which is a combination of a weighting and the geometric median, to understand how it performs under a variety of different noisy perturbations. We compare its robustness against a model that uses a weighted mean instead of a weighted geometric median and find that our approach is significantly more robust (see Supplementary Fig. 2). Finally, the (weighted) geometric median is translation equivariant and orthogonal equivariant[50] which means that the relationship between the bands is maintained under orthogonal transformations and shifts (a mathematical fact that follows

easily due to the invariance of Euclidean norm under such transformations). This is not true, for example, if the coordinatewise median is used (i.e., a univariate median in each band separately). This final property is particularly important for machine learning due to the extensive use of Principal Component Analysis (PCA) in the analysis of satellite imagery as PCA is a statistical and data reduction procedure that uses an orthogonal transformation to convert observations of possibly correlated variables into a set of values of linearly uncorrelated variables called principal components.

## Data availability

Both the Landsat-8 Barest Earth mosaic and the Landsat 30+ mosaic are freely available through the web viewer https://nationalmap.gov.au. Alternatively, these mosaics (sized around 200GB each, in Australian Albers projection, tiled, and in GeoTIFF format) can be obtained by contacting Geoscience Australia.

## Code availability

The code used to generate the Barest Earth mosaics from freely available Landsat data and the code to validate results against the spectra of the soil samples collected in the field is available through the GitHub repository of the first author: https://github.com/daleroberts

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

## Acknowledgements

D.R. and J.W. would like to thank Silvio Mezzomo, Chris Evenden and David Arnold at Geoscience Australia (GA) for graphical support, Richard Blewett (GA), Karol Czarnota (GA), Norman Mueller (GA), and Simon Oliver (GA), Geoff Fraser (GA) and Alexis McIntyre for scientific discussions and support, Patrice de Caritat (GA) and Ian Lau for the NGSA site data and spectra. GA's Digital Earth Australia team for their ongoing infrastructure support and (in particular) David Gavin and Harshu Rampur for preparing and hosting the products on nationalmap.gov.au. This paper is published with the permission of the CEO, Geoscience Australia. The research is a product of the 'Exploring for the Future' project.

## Author contributions

J.W. posed the problem of obtaining a high-quality barest Earth mosaic of Australia. D.R. designed the mathematical approach to solve the problem, wrote the codes, ran the codes on the continental-scale satellite imagery dataset and performed the validation against the NGSA dataset. J.W. and D.R. jointly contributed to figure preparation, results interpretation and writing the final manuscript. O.G. helped with data collection for test areas, preparation of some figures and evaluating various weighting schemes over the test areas.

## Competing interests

The authors declare no competing interests.
