## [Peer Review File · Nature Communications]

**REPORT ON THE MANUSCRIPT
“REVEALING THE AUSTRALIAN CONTINENT AT ITS BAREST”**

SUBMITTED TO NATURE COMMUNICATIONS

The goal of the work is to use a large Landsat dataset to extract the type surface materials (soil and rock) from the measurements that are contaminated by noise due to vegetation covering the surface. This is made possible due to availability of multiple measurements (some of which could be less affected by vegetation) for the same surface region over time. Authors apply the mathematical method based on the *weighted geometric median* to construct a “robust” estimate of the “barest state” (i.e. least affected by vegetation).

I am not an expert in surface mineralogy and my main goal was to assess the feasibility and soundness of proposed mathematical model. The geometric median of a collection of points estimates the “centroid” of the underlying data-generating distribution defined in the following way: in a company where employees can be modeled as “points,” the geometric median describes the location of the office that *minimizes the total distance traveled by the employees to get to work* (whereas the mean minimizes the sum of *squared* distances traveled). Robustness in this case can be characterized by the property that the relocation of a small number of employees does not require the office to be moved as well. Translated to the framework of the paper, when the median is used to estimate the centroid of the noisy surface measurements, it will “ignore” (more precisely, downweight) the measurements at times when the vegetation was very high or very low compared to a “typical” state. The problem here is that the measurements with very low vegetation level are the most valuable ones, so they should receive highest weight, not lowest. Authors address this issue by explicitly adding data-dependent weights to observations, with a weighting scheme aiming to attach largest weights to observations with least amount of vegetation, as the intuition suggests. *My opinion is that the weighting scheme is the main ingredient of the method, and its impact on the final result is at least as large as the effect of chosen aggregation method (e.g., mean or the median).* With this in mind, I would like to see the following questions addressed:

- (a) Authors state that “A key point is that by weighting (as opposed to filtering) we do not throw away any information.” This needs to be explained better. My intuition (that could be wrong) is the following: there are two types of “noise,” first is the bias due to vegetation and second is noise due to measurement errors, pixel corruption, etc. To remove the first type of noise, the best approach is to take the observation with the least amount of vegetation, and ignore the rest. To remove noise of the second type, it is best to average all of the data (in case there are no outliers) or to use the (non-weighted) geometric median. So, the good weighting scheme should “balance” these two ideas.
- (b) The choice of weights appeared to be somewhat arbitrary is not well-explained. What was the authors’ logic behind the choice? It is possible that there are certain “standard” ways to assign these weights, but the construction in the paper appeared to be not well-motivated, at least to a non-expert like myself. One point to keep in mind is that, in order for the method to be robust, the data-dependent weights themselves should be robust in the proper sense

(for instance, the point with high vegetation should never receive large weight). Authors should discuss this question in relation the weights used in the paper.

- (c) I wonder whether authors tried to compare weighted geometric median with the weighted average – if the former produces better results, it would be a strong argument in favor of robust techniques.
- (d) Authors mention that “..the approach does not use any spatial information.” What is the justification here? Is it not feasible computationally, or does it not produce noticeable improvements?
- (e) Authors state that “Due to the high-dimensional nature of our technique, the spectral integrity (e.g. correlations between bands) is maintained across all wavelengths. ” I did not understand how the dimensionality of the data is related to preservation of correlations. It is also stated in the abstract that “Importantly, our method preserves the spectral relationships between different wavelengths of the spectra. ” This claim requires more formal explanation: is it a mathematically justified fact, or just an empirical observation?
- (f) It is also possible that results could be further improved via considering the *weighted Huber’s loss* instead of the WGM, defined as follows: let $H(x) = \begin{cases} x^2/2, & |x| \leq 1, \\ |x| - \frac{1}{2}, & |x| > 1, \end{cases}$ be

Huber’s loss. Now define

$$m_w := \operatorname{argmin}_{x \in \mathbb{R}^p} \sum_{t=1}^n w_t H \left(\frac{\|x - x^{(t)}\|}{\Delta} \right),$$

where w_t ’s are the appropriate weights and the (user-set) parameter $\Delta > 0$ balances “robustness” and efficiency, meaning that when $\Delta \rightarrow 0$, we get the geometric median and when $\Delta \rightarrow \infty$, we get the average that is better if there are no outliers/grossly corrupted data. I don’t expect that authors will implement this approach in the paper under consideration, but I hope they will try it in the future.

Minor remarks:

- (1) Page 3, [qiu2017developing] appears to be a typo in place of the citation number.

Overall, I enjoyed reading the paper, and my opinion is that it presents an interesting approach to a challenging problem.

Reviewer #2 (Remarks to the Author):

The paper presents an interesting statistical technique to generate a continental scale spectral response of the bare surface based on the Landsat data. The reasoning of the techniques reads logical. The potential applications of the spectral products are broad if the technique truly works as the authors argued. However, the authors need to improve the description of the methods, enhance the validation steps, and clarify the key points of the result discussion. The paper needs a moderate revision before it can be accepted for publication.

1. The presentation of the methods was not clear.

There are two concepts involved with the WGM (weighted geometric median): 1) what is the dimension of the geometric median, the bands, time-series, image pixels or some types of combinations? 2) what did you want to weight, the bands, time-series, or image pixels?

The description of the methods and the fitting of the methods were not consistent, or at least not conforming well.

The equations were not numbered in the manuscript, but were referenced by the numbers, which should be corrected.

2. The validation steps need to be enhanced.

“(Figure 4). Focusing on specific pixels within this area, we show in Figure 5 the variation in spectral responses and our WGM estimate of the barest state.” Figure 5 is the key evidence to support the second validation. However, it was referred in the first validation, a small test area, which was not consistent with the 1st validation, but repeated the 2nd validation.

The description of the third validation was too brief, from which we couldn't see what the authors planned to tell.

3. The discussion of results needs to be closely related to the main theme of the paper.

The current discussions were focused on the challenges and limitations of the proposed new statistical technique. Based on this section, it reads like that the new method was not effective, but severely limited.

4. Inconsistencies were found in several places, for instance, no Figure 6a was labeled in the figures, but was referred in the text. The authors should proofread the manuscript before submitting it.

Yichun Xie

RESPONSE TO REVIEWERS

RE: Revealing the Australian Continent at its Barest.

We thank the reviewers for their careful reading of our manuscript, their constructive comments, and their opinion that our paper has considerable potential interest to a broad audience. As such, we have systematically addressed all the comments and concerns from the review and are now proposing the attached revised version of the manuscript to Nature Communications for consideration.

We highlight that compared to our original paper:

- Based on Reviewer 2's comment about validation, we have completed a substantial third validation of our mosaic whereby we have compared the 25m² spectra obtained from our model to soil samples collected in the field by an independent organisation CSIRO. See below.
- Even though Reviewer 1 mentions "*I don't expect that authors will implement this approach in the paper under consideration, but I hope they will try it in the future*", we have extensively explored their suggestion of using a Huber loss approach instead of the geometric median approach that we use for our method (see below). It was a useful exercise to ensure that our results are optimal in the context of satellite imagery.

Reviewer #2 (Yichun Xie):

The paper presents an interesting statistical technique to generate a continental scale spectral response of the bare surface based on the Landsat data. The reasoning of the techniques reads logical. The potential applications of the spectral products are broad if the technique truly works as the authors argued. However, the authors need to improve the description of the methods, enhance the validation steps, and clarify the key points of the result discussion. The paper needs a moderate revision before it can be accepted for publication.

The presentation of the methods was not clear. There are two concepts involved with the WGM (weighted geometric median): 1) what is the dimension of the geometric median, the bands, time-series, image pixels or some types of combinations? 2) what did you want to weight, the bands, time-series, or image pixels?

In our original submission, we originally explained this in the DATA AND APPROACH section as:

"For each time series, our aim is to obtain an estimate of the barest state observed relating to either soil or exposed rock. To achieve this we holistically consider each pixel observation through time as 6-dimensional vector."

Based on reviewer feedback, we have revised this sentence to:

"Our aim is to obtain an estimate of the barest state observed relating to either soil or exposed rock by using the time series of observations. To achieve this we consider each pixel separately and take its 6-band time series as the sample observations of this location."

We compute a weighted geometric median of these 6 dimensional observations where the weights penalise vegetated observations and favour bare observations. This effectively preferences the bare Earth spectral signature over heterogeneous spectral responses through the time series related to vegetation, atmospheric noise, and data corruption (e.g., clouds, shadows, saturation)."

We have also reviewed our METHODS section to ensure that all details are given so that the algorithm can be replicated by others. For example, the dimensionality ($p=6$, i.e. the number of bands) and how the pixels are weighted through time (based on relative bareness). In addition, we will make the source code freely available on GitHub once the paper is accepted to ensure that our results are reproducible by others and that the algorithm details are explicit.

The description of the methods and the fitting of the methods were not consistent, or at least not conforming well.

We have addressed this comment by adding a summary of our approach in the main part of the paper:

"The weighting scheme was derived by fitting an ensemble of regression models based on spectral features over various training areas exhibiting a high variability of biophysical states. A single model was derived from this ensemble using a loss function that minimises NDVI and variability over these areas; see METHODS."

We provide exact details of our approach in the METHODS 'Fitting an optimal model' section, however based on reviewer feedback we have attempted to more clearly articulate our approach here by adding a summary paragraph before we delve into the details:

"To reduce the potential of overfitting our weight model, we propose an approach that can viewed as fitting an ensemble of regression models based on spectral features from which we derive a single model from this ensemble through a loss function that minimises average NDVI and variability over our training sites. We now describe this more precisely."

The equations were not numbered in the manuscript, but were referenced by the numbers, which should be corrected.

We faced limitations using Word (instead of LaTeX) in the placement of the numbering with the view that the numbering location would be placed to the right of the equation by the journal typesetter. For example, we point to our original manuscript p.3, where we number Eq.1:

"...features have the form (Eq. 1):"

We have updated this to "[Eq. 1]" so that it is more visible to the journal typesetter.

And Eq.2:

"... define our (unnormalised) weight function as (Eq. 2):"

is now "[Eq. 2]".

2. The validation steps need to be enhanced. "(Figure 4). Focusing on specific pixels within this area, we show in Figure 5 the variation in spectral responses and our WGM estimate of

the barest state.” Figure 5 is the key evidence to support the second validation. However, it was referred in the first validation, a small test area, which was not consistent with the 1st validation, but repeated the 2nd validation.

We thank the reviewer for spotting this and acknowledge a mistake in the figure numbering, we have corrected the text to:

“... period (Figure 3). Focusing on specific pixels within this area, we show in Figure 4 the variation in spectral responses and our WGM estimate of the barest state.”

And we have also revised numbering within the caption of Figure 4. This should address the concerns related to this part of the validation process.

The description of the third validation was too brief, from which we couldn't see what the authors planned to tell.

We thank the reviewer for bringing this to our attention. Based on this feedback, we have significantly revised our discussion on the third validation and have performed a more extensive quantitative validation to summarise the way in which our Barest Earth model reduces the influence of vegetation and enhances the match to the mineralogy (mineral spectra) of the top 10cm of soil. This process is carried out using ground truth measurements collected across the Australian continent. We have included two new Figures that illustrate the significance of our results by comparing the Barest Earth spectra and the soil spectra versus a Landsat observation with vegetation present (VEG).

We have added a number of new Figures to show the “ground truth” validation of our results; see Figure 6 and 7.

3. The discussion of results needs to be closely related to the main theme of the paper. The current discussions were focused on the challenges and limitations of the proposed new

statistical technique. Based on this section, it reads like that the new method was not effective, but severely limited.

Based on this comment, we have rewritten our Discussion section to make it clearer.

4. Inconsistencies were found in several places, for instance, no Figure 6a was labeled in the figures, but was referred in the text. The authors should proofread the manuscript before submitting it.

We thank the reviewer for their careful reading and bringing these errors to our attention. We have proof-read our manuscript to ensure that all Figure labels are correct (please also see comments above).

Reviewer #1:

The goal of the work is to use a large Landsat dataset to extract the type surface materials (soil and rock) from the measurements that are contaminated by noise due to vegetation covering the surface. This is made possible due to availability of multiple measurements (some of which could be less affected by vegetation) for the same surface region over time. Authors apply the mathematical method based on the *weighted geometric median* to construct a “robust” estimate of the “barest state” (i.e. least affected by vegetation).

I am not an expert in surface mineralogy and my main goal was to assess the feasibility and soundness of proposed mathematical model. The geometric median of a collection of points estimates the “centroid” of the underlying data-generating distribution defined in the following way: in a company where employees can be modeled as “points,” the geometric median describes the location of the office that *minimizes the total distance traveled by the employees to get to work* (whereas the mean minimizes the sum of *squared* distances traveled). Robustness in this case can be characterized by the property that the relocation of a small number of employees does not require the office to be moved as well. Translated to the framework of the paper, when the median is used to estimate the centroid of the noisy surface measurements, it will “ignore” (more precisely, downweight) the measurements at times when the vegetation was very high or very low compared to a “typical” state. The problem here is that the measurements with very low vegetation level are the most valuable ones, so they should receive highest weight, not lowest.

We agree with the reviewer, the measurements with low vegetation are the most valuable ones so they receive the highest weight and the measurements with high vegetation receive the lowest weight. We originally summarised this on page 2:

“Our motivation to include weights into the definition of the GM, giving the WGM, is to focus the WGM on observations through time that exhibit the least amount of vegetation...”

Based on reviewer feedback we have attempted to sign-post this better in our paper; see revisions made based on Reviewer 2’s comments (above).

Authors address this issue by explicitly adding data-dependent weights to observations, with a weighting scheme aiming to attach largest weights to observations with least amount of vegetation, as the intuition suggests. My opinion is that the weighting scheme is the main ingredient of the method, and its impact on the final result is at least as large as the effect of chosen aggregation method (e.g., mean or the median).

We agree with the reviewer: the novelty in our methodology is not only the use of a weighted geometric median but also the development of a scheme to assign weights to observations, and finally the fitting of this weighting scheme.

The ‘weighted geometric median’ as the aggregation method is also important to the success of our approach due to its high breakdown point (50%) compared to the breakdown point of the mean (0%) and the multidimensional nature of this robust method (compared to a coordinate-wise univariate median) so that the resulting estimate of the bare spectra maintains the correct relationships between the bands. This relationship is important if the bare spectra are to be further analysed using principal component analysis and other machine learning techniques.

With this in mind, I would like to see the following questions addressed:

Authors state that “A key point is that by weighting (as opposed to filtering) we do not throw away any information.” This needs to be explained better.

Due to the Petabyte-sized dataset that we are dealing with and the heterogeneity of this data there are a number of downsides to filtering.

First, filtering implies that we need to have a classification model of what a “non-bare” pixel looks like in **all** possible cases and then filter those away, however, over the whole continent of Australia this would be nearly impossible to do perfectly. Some areas would be incorrectly filtered leading to outliers in the data which, in turn, could lead to incorrect estimation of the spectra if one uses, for example, the mean of the observations (we comment on this later).

Second, there are areas across the continent that are never bare. For example, permanent forest. This means that if these observations are filtered (assuming you could do this perfectly) then there would be locations with no observations which would lead to a mosaic with missing data. We feel that this is not an optimal outcome as our Barest Earth mosaic is designed specifically to be used by other models (e.g., spatial regression) to understand relationships between bareness and other covariates across the continent.

To show the effect of filtering, we compare our results to those of Rogge et al. (2018) to demonstrate the significant effect this plays.

Figure 10 from Rogge et al. 2018 showing an example composite image (Landsat bands 7-5-3) using their filtering method exhibiting missing values in output (black) over Germany.

Above is a section of our Barest Earth mosaic over Australia for a similar sized region to Germany and using the same visualisation (Landsat bands 7-5-3). Our result shows no missing values (black areas in Rogge et al. 2018) in output and with a significantly improved visual and spectral result compared to Rogge et al. (2018). Location is the 'Great Australian Bight'. The black at the southern part of the image is ocean.

My intuition (that could be wrong) is the following: there are two types of “noise,” first is the bias due to vegetation and second is noise due to measurement errors, pixel corruption, etc. To remove the first type of noise, the best approach is to take the observation with the least amount of vegetation, and ignore the rest. To remove noise of the second type, it is best to average all of the data (in case there are no outliers) or to use the (non-weighted) geometric median. So, the good weighting scheme should “balance” these two ideas.

We believe that “*take the observation with the least amount of vegetation, and ignore the rest*” is equivalent to filtering. As we mentioned above, this leads to the situation where there will be locations (pixels) with no observations resulting in a model output with missing data.

We agree with the reviewer that averaging all of the data makes sense when there are no outliers. However, a peculiarity of satellite imagery is that it often has more outliers than clear observations. This means that nearly every single one of the billions of time series across the continent will experience one or multiple large outliers through time (e.g., thick cloud, sensor saturation), multiple smaller outliers (thin cloud, shadows, etc.), and atmospheric noise. Often, we find ourselves in the situation where the number of clear observations n is equal to (or less than) the dimension of observation vector p .

We recall that the breakdown point of the mean is 0% while the median and geometric median have breakdown points of 50%. Roughly, the breakdown point is the percentage of outliers in the data that the statistic can handle before giving incorrect results; see Lopuhaä and P. J. Rousseeuw (1991) for a more precise mathematical statement.

As satellite imagery always has a significant amount of outliers and due to the 0% breakdown point of the mean estimator, we feel that using the mean estimator is not the best approach.

In turn, any ‘balancing’ which mathematically can be viewed as a linear combination of a mean estimator and median estimator will have a breakdown point lower than 50% which is not optimal. Therefore, we feel that our geometric median estimator will always give a better result for satellite imagery compared to an estimator that balances between a mean and a median. We show this in practice below where we have performed a full study to compare the effect of using a mean instead of a median.

Finally, choosing a single estimator (rather than a mix of two) is easier to explain to end users of the model output. This is significant from a policy perspective as Geoscience Australia will distribute the results freely to the public.

We have added the following reference to the paper:

Breakdown points of affine equivariant estimators of multivariate location and covariance matrices. (1991). H. P. Lopuhaä and P. J. Rousseeuw. *Annals of Statistics*, Vol. 19, No. 1, pp. 229-248.

The choice of weights appeared to be somewhat arbitrary is not well-explained. What was the authors’ logic behind the choice? It is possible that there are certain “standard” ways to assign these weights, but the construction in the paper appeared to be not well-motivated, at least to a non-expert like myself.

The ‘features’ (aka. covariates, predictor variables) in our weight model are based on standard normalised difference ratios of the bands. The simplest of these ratios take the form $(a-b)/(a+b)$ for bands/channels a and b , examples include NDVI and NDWI; see [Eq 1] in paper and discussion after. This approach is classic in the remote sensing literature and, as such, is one of the reasons that we use it as it improves interpretability. Further, they are

robust to atmospheric effects which improves the overall robustness of our approach; we discuss this further below.

Our final Bare Earth model for the whole continent uses NDVI but we give a more general form in our paper as we explore many models and find that $-3 \times \text{NDVI}$ gives the optimal one for our loss function (see Figure M2). We settled on this weight / feature as it is simply a scaled version of a classic indicator of bareness in the remote sensing literature. However, our methodology is general and can accommodate linear combinations of these types of features such as those found in various papers (as we point out in the paper in the discussion after [Eq 1]). Our model search (ensemble regression) found that having this single weight was optimal in that it did not overfit any particular region of the Australian continent.

One point to keep in mind is that, in order for the method to be robust, the data-dependent weights themselves should be robust in the proper sense (for instance, the point with high vegetation should never receive large weight). Authors should discuss this question in relation to the weights used in the paper.

The model features used to determine the weights are simply an abstraction of classic ‘band ratios’ that are used in remote sensing. Normalised difference band ratios are used in remote sensing due to their well-known robustness to atmospheric effects based on the theory that atmospheric noise affects all bands/channels in an equal manner, giving “white” clouds in the most extreme case. As such, it is easy to see that for some perturbation ϵ across all bands we have $((1 + \epsilon) a - (1 + \epsilon) b) / ((1 + \epsilon) a + (1 + \epsilon) b) = (a - b) / (a + b)$. In other words, NDVI and other normalised difference band ratios are robust to parallel shifts across all the wavelengths, i.e., invariant to “albedo shifts”. This means that our weight model is robust in that sense.

We have also added a section on model robustness in the METHODS section of the paper to address this comment.

I wonder whether authors tried to compare weighted geometric median with the weighted average – if the former produces better results, it would be a strong argument in favor of robust techniques.

The usage of the Weighted Geometric Median (WGM) in this work was in fact motivated by the failure of the weighted average to adequately capture the central tendency of raw Landsat data. As alluded to in the response to the first comment, the median is a robust estimator, whilst the average is not. Existing approaches, such as Rogge et al. (2018) described above, that make use of the weighted average to summarize the value of a pixel through time must first go through a preprocessing step in which outliers are removed before taking an average of the remaining inlier points. Given that RSI data exhibits a high proportion of corrupted observations, such a preprocessing step is quite costly and often ineffective. For example, filtering can be performed by building a classifier to detect extremes, or by setting hard thresholds for outliers, neither of which is foolproof. In contrast, the median allows us to bypass this preprocessing step altogether and achieve a valid estimator for the centroid of the data.

Our conclusion is that robust techniques are demonstrably superior in this domain. In order to illustrate this, we performed an empirical study to demonstrate the performance of our robust technique by comparing the output of our geometric median approach with the outputs of an averaging approach, and a Huber’s loss approach (suggested in a later comment). The results are as follows:

The first column shows the NIR band of a pixel composite using the three methods (rows) based on data that has no cloud masking applied. The second row shows the same with data where cloud masking has been applied. The third column shows the NIR band of a single clear observation (replicated three times down the column). The fourth column gives the result of column two minus column three.

It is clear from this study that the mean (i.e., averaging) is not robust and does not give a good result on satellite imagery due to the amount of outliers present (see column 1) even if cloud is removed as much as possible (see column 2). The Huber loss with a positive delta is also not robust enough (see row 3). In conclusion, we see that our approach is the most optimal for our situation.

Note: We have decided not to place this further study suggested by the referee in the final paper but would be happy to change this if the Editor believes it would add to the paper.

Authors mention that “..the approach does not use any spatial information.” What is the justification here? Is it not feasible computationally, or does it not produce noticeable improvements?

Initially we started from the concept that bareness is not a phenomena influenced by how a locations neighbours are bare but is largely a temporal concept. For example, there could be a dense forest that never becomes bare right next to a cropping area that does. This is why we call it ‘Barest Earth’ and not ‘Bare Earth’ as we are estimating the barest spectra we can observe through time.

There are also some other reasons:

1. A spatial model needs to take into account both outliers (e.g., clouds / shadows) and missing values (e.g., edge of scene) in neighbouring pixel locations. This leads to significant complexity in the model.
2. As our weighting scheme can be viewed as a ranking of bareness throughout time (as opposed to a classification), introducing spatial information makes it difficult to design an ordering/ranking. An analogy can be made to attempting to find a natural linear ordering of the complex numbers.
3. One of the aims of introducing spatial information is that it may increase the spatial regularity of the output. However, it has been shown in the paper ‘Roberts et al. (2018)’, the geometric median pixel composite mosaic methodology provides an output that has a high amount of spatial regularity already.

That said, the cloud/shadow masking machine learning algorithm that is used to remove as much cloud and shadow possible, uses spatial and also other ancillary information to classify.

Authors state that “Due to the high-dimensional nature of our technique, the spectral integrity (e.g. correlations between bands) is maintained across all wavelengths. ” I did not understand how the dimensionality of the data is related to preservation of correlations. It is also stated in the abstract that “Importantly, our method preserves the spectral relationships between different wavelengths of the spectra. ”. This claim requires more formal explanation: is it a mathematically justified fact, or just an empirical observation?

Our comment here in regards to the preservation of correlations is to distinguish our approach to other multivariate robust methods such as taking coordinatewise medians in each band separately. This is due to the mathematical fact that the geometric median is translation equivariant and orthogonal equivariant (as opposed to the coordinatewise median); see Lopuhaä and P. J. Rousseeuw (1991) for the mathematical argument.

First, to show our point visually, we have generated the following study comparing the robustness of the geometric median, the mean, and the coordinate median (i.e., univariate median in each band separately). From this study, it is clear that the coordinate median is not robust to factorising out the correlation.

Another way of saying this is that the relationships between the bands does not depend on the coordinate system of the data and, if desired, the relationship between bands can be factored out of the Barest Earth estimate by taking the singular value decomposition (SVD) of the data and factoring out the orthogonal matrices.

Mathematically, this can be described as follows: If X is a p -dimensional random vector modelling the radiance at a particular pixel (i.e., the radiance observations are sampled from this distribution). Then without loss of generality, it can be written as $X = UZ + v$ where Z is a random variable such that $E[Z] = 0$, $cov(Z) = S = diag(s_1, \dots, s_p)$ and U is an orthogonal matrix. We note this claim follows easily by de-meaning taking the singular value decomposition of $cov(X)$. As the geometric median is orthogonal equivariant, it follows that if $m(\cdot)$ is the geometric median then $cor(m(X)) = cor(A m(Z) + v) = A cor(m(Z))A^T = A A^T$ and $cor(X) = A cor(Z)A^T = AA^T$.

We note that, from an application perspective, we are not necessarily interested in obtaining a good estimate of the correlations/covariances at one particular spatial location (ie. pixel) but across all spatial locations. In Roberts et al. 2018, we showed that the geometric median reduces noise spatially which improves the estimation of these correlations. This, in turn, is useful for performing machine learning on our Barest Earth mosaic.

It is also possible that results could be further improved via considering the *weighted Huber's loss* instead of the WGM, (...) where w_t 's are the appropriate weights and the (user-set) parameter $\Delta > 0$ balances "robustness" and efficiency, meaning that when $\Delta \rightarrow 0$, we get the geometric median and when $\Delta \rightarrow \infty$, we get the average that is better if there are no outliers/grossly corrupted data. I don't expect that authors will implement this approach in the paper under consideration, but I hope they will try it in the future.

We agree with the reviewer that usage of Huber's loss generalizes our approach, however (see above), we performed an experimental study of Huber's loss and compared the output against our WGM technique and a technique based on taking averages. Although it is indeed a generalisation of our approach, it has difficulties that may limit the usefulness of such a generalization.

First, when working with Landsat data, a high proportion of the observations (pixels through time) are prone to missing values, corruption and measurement error, as well as other noise such as cloud cover – we would therefore expect that using Huber's loss would act very similarly to the geometric median, that is to say, we would expect $\Delta \approx 0$ in any approach in which Δ is estimated from the data.

Second, the introduction of an additional hyper-parameter Δ introduces an extra layer of complexity to the methodology, given that a flexible and unbiased framework should make use of a data-driven approach for estimating/learning Δ , for example: cross validation. Further, it is not quite clear if a single Δ can be found for the whole 16 billion time series required to cover the continent of Australia or if each of these time series requires a different Δ that may depend on the epoch that the model is considered over, the amount of cloud coverage, etc.

In conclusion, we feel that our WGM is optimal in the context of satellite imagery.

**REPORT ON THE REVISED MANUSCRIPT
“REVEALING THE AUSTRALIAN CONTINENT AT ITS BAREST”**

SUBMITTED TO NATURE COMMUNICATIONS

The revision is a definitely an improvement over the first version of the manuscript. I want to thank the Authors for carefully addressing my comments and providing detailed response. The algorithms are now better motivated and explained.

My only remaining concern is related to clarifying robustness. Specifically, it is stated on page 4 that “our weight function is also robust.” This point requires some justification. I find the explanation in the response to be insufficient as it considers only parallel shifts, a very specific form of perturbation across the bands (unless other types of corruption are not possible). The issue is not straightforward as the weights are data-dependent, and change for various datasets. To see why this is an issue, note that the sample mean \bar{x}_n of points $x_1, \dots, x_n \in \mathbb{R}^p$ is

$$\bar{x}_n = \operatorname{argmin}_{z \in \mathbb{R}^p} \sum_{j=1}^n w_j \|z - x_j\|$$

where $w_j = \|x_j - \bar{x}_n\|$, so robustness of the weights can’t be “separated” from robustness of the estimator itself. It was not immediately clear from the weight selection algorithm why the weights will be robust; while a complete mathematical treatment of this point is not necessary (unless authors decide that it can be done), a convincing numerical example would be beneficial. Such an example should include a plot showing the evolution of the “optimal weights” as more outliers of large magnitude (for instance, randomly generated vectors from normal distribution with large mean and covariance) are added to the dataset. Alternatively, an experiment can consist in adding artificial contamination to the dataset and looking at the outputs of the algorithm as a whole (instead of just the weights). I leave the exact design of the experiment to the authors.

Minor remarks:

- (1) Page 5, “..propose an approach that can viewed” should be “propose an approach that can BE viewed.”

Reviewer #2 (Remarks to the Author):

I read the revision with a good interest. The authors have addressed my comments point-by-point. I am happy to say that the points raised in the previous round of review have been satisfactorily addressed.

Reponse to referees

We thank the referees for their careful reading of our manuscript and address the final query from one of the referees concerning the robustness *with respect to the proposed weight function* of our algorithm for estimating the barest spectra at every location. The referee suggested that “the explanation in the response to be insufficient as it considers only parallel shifts, a very specific form of perturbation across the bands (unless other types of corruption are not possible).” As such, we have now performed a more extensive study of our algorithm for a wider variety of perturbations. Our key findings are:

- Our proposed algorithm is robust against a variety of different corruptions.
- Our approach is significantly more robust than using a ‘weighted mean’ as the summary statistic.

As such, we feel that we have adequately addressed the referee’s comment (in the positive). These claims are demonstrated below in detail with a series of simulation studies, including the code to allow the reviewer to replicate our study if required.

We have also corrected a minor typo, as pointed out by the reviewer, and made some additions to the methods part of the paper to include a summary of the simulation study results.

INTRODUCTION

We believe that the referee suggests that we have three types of outliers (or corruption) in our study:

- (1) *Thick cloud* that could possibly give a non-parallel spectral shift to pixel. We note that these are masked (and set as missing data) using the FMASK algorithm (Zhu & Woodcock, 2014) which is a machine learning / physical algorithm with high accuracy on this type of corruption.
- (2) *Thin cloud / atmospheric disturbances / shadows* which are often assumed (in the remote sensing literature) to only given by parallel shifts of the spectra as they typically only change the albedo (brightness / whiteness) of the surface reflectance.
- (3) An observation giving a bare signal but that is not bare (e.g., exhibiting a low NDVI signal but that is not bare). This form of corruption would be a spectral change that is not just a parallel shift.

We will now address these forms of corruption and noise through various simulation studies. We follow the approach suggested by the referee: “the experiment can consist in adding artificial contamination to the dataset and looking at the outputs of the algorithm as a whole (instead of just the weights)”. We also perform the same experiment against the *weighted mean* as a comparison. In particular, our aim is to understand the influence of contaminants on the final output as the percentage of contaminants increases up to the theoretical breakdown point of the geometric median (50%).

We first note that our observations in our study are measures of surface reflectance. This puts us in a special situation, in that, each vector observation $\mathbf{x} = (x_1, \dots, x_p)'$ must satisfy $x_i \in [0, 1]$ for $i = 1, \dots, p$ as surface reflectances is a percentage. This actually makes a simulation study to validate our claims a little more difficult than simulating from a multivariate normal with a specific mean μ and covariance Σ as we need to ensure that values fall in the range $[0, 1]$. Therefore, we will assume that:

- $p = 6$ with coordinate labels BLUE, GREEN, RED, NIR, SWIR1, SWIR2.
- Our observations are sampled $\mathbf{x} \sim \text{TruncatedMVN}(\mu, \Sigma; 0, 1)$, i.e., sampled from a *truncated* multivariate normal where the coordinates are bounded between 0 and 1.

This response will include the full **R** code so that all our workings are clear, transparent, and the referees can reproduce our robustness study if they desire.

SETUP

We will use the following package to draw realistic samples bounded between $[0, 1]$:

```
library(TruncatedNormal)
```

and we also set the dimensionality of our observations as

```
p <- 6
```

Construction of our weighting model. Although we give a general methodology for choosing a weighting function in our paper, we shall restrict ourselves in this simulation study to the specific choice of weighting model that was used to produce our final output products.

At the core of our weighting function, we used the *normalised difference vegetation index* (NDVI) as a measure of greenness in the landscape. It has the form $(a - b)/(a + b)$ where a is the NIR band and b is the RED band. We made this choice as it is well understood in the area of remote sensing and it also has the following interesting mathematical property:

$$\frac{(a(1 + \varepsilon) - b(1 + \varepsilon))}{(a(1 + \varepsilon) + b(1 + \varepsilon))} = \frac{(a - b)}{(a + b)}.$$

This means that this index is robust to parallel shifts and it is also constrained between $[-1, 1]$. It is well-known in the remote sensing literature that high values of NDVI indicate the presence of vegetation due to various biophysical arguments about photosynthesis.

Next, we introduce a parameter $\nu > 0$, and multiplied NDVI by $-\nu$ to inverse the biophysical relationship so that higher values give the absence of vegetation. Based on minimising an objective function over various test areas (see the training section in our paper), our final choice of ν was:

```
nu <- c(3)
```

Finally, we rescale all the weights of the observations through time using a SOFTMAX. This gives the following (vectorised) function for computing our weights:

```
weight.softmax <- function(X, nu=3) {
  w <- nu * -1 * (X[,4] - X[,3]) / (X[,4] + X[,3])
  w <- exp(w)
  w <- w / sum(w)
  w[is.nan(w)] <- 0
  return(w)
}
```

An alternative simple weighting is to transform -NDVI to lie in the range $[0, 1]$ by shifting by 1 and dividing by 2.

```
weight.simple <- function(X) {
  w <- (-1 * (X[,4] - X[,3]) / (X[,4] + X[,3]) + 1) / 2
  w[is.nan(w)] <- 0
  return(w)
}
```

The weighted geometric median and weighted mean algorithms. In this simulation study, we will use the Weiszfeld algorithm to compute the weighted geometric median. We shall use the implementation from the following package:

```
library(Gmedian)
```

We wrap the function provided by the **Gmedian** package to deal with the edge case where all the weights may possibly be zero. We also set some default parameters for the algorithm.

```
geomedian <- function(x, w=c(0)) {
  if (sqrt(sum(w^2)) < 1e-9) {
    # All weights zero, so assume all observations have same weight 1/n
    gm <- Weiszfeld(x, epsilon=1e-9, nitermax=10000)$median
  } else {
    gm <- Weiszfeld(x, weights = w, epsilon=1e-9, nitermax=10000)$median
  }
}
```

```

    return(as.numeric(gm))
}

```

We also introduce a function to compute the weighted mean of vector data for comparison to our approach (as one of the referees raised this the last review round).

```

wmean <- function(X, w) {
  if (sqrt(sum(w^2)) < 1e-9) {
    return(apply(X, 2, function(x) mean(x, na.rm=TRUE)))
  } else {
    return(apply(X, 2, function(x) weighted.mean(x, w, na.rm=TRUE)))
  }
}

```

Additional utility functions. We introduce a few utility functions that we shall use in the study. For example, we will plot observations using a parallel plot.

```

parplot <- function(X, ...) {
  matplot(1L:ncol(X), t(X), type = "l", lty=1, xaxt = "n", xlab="",
    ylab="", ylim=c(0,1), ...)
}

```

We use this package to set the colors for our plots.

```

library(scales)

```

We define the cosine and Euclidean distance functions for evaluation of closeness between vectors.

```

euclidean <- function(x, y) {
  sqrt(sum((x - y)^2))
}

cosine <- function(x, y) {
  denom <- sqrt(sum(x^2)) * sqrt(sum(y^2))
  as.numeric(1 - (t(y) %*% x) / denom)
}

```

We define a clipping function to force values to stay within $[a, b]$ if required.

```

clip <- function(x, a, b) {
  a + (x-a > 0)*(x-a) - (x-b > 0)*(x-b)
}

```

The simulation study function. We define a function `sim.study` that performs a simulation study given a specific corrupter function and parameters for the distribution that the observations are sampled from (i.e., μ and Σ). The corrupter function, that must be defined for each case that we investigate, takes a vector observation and corrupts it in some way.

```

sim.study <- function(corruptor, mu, Sigma, n.sim=100, n.obs=1000) {
  par(mgp = c(3, 0.2, 0), tck=-0.01) -> opar

  # corruption percentage
  pc <- c(seq(0.01, 0.05, by=0.01),
    seq(0.1, 0.4, by=0.1),
    seq(0.45, 0.49, by=0.01))

  wmd <- matrix(NA, ncol=length(pc), nrow=n.sim)
  gmd <- matrix(NA, ncol=length(pc), nrow=n.sim)

  plot(pc, pc, type='n', ylim=c(0, 0.6), xlab="", ylab="")
}

```

```

for (k in 1:n.sim) {

  X <- rtmvnorm(n.obs, mu, Sigma, lb=rep(0, p), ub=rep(1, p))

  gm <- geomedian(X, weight.softmax(X))
  wm <- wmean(X, weight.softmax(X))

  for (i in 1:length(pc)) {
    m <- floor(pc[i] * n.obs)
    cX <- matrix(X, nrow=nrow(X))
    idx <- sample(1:n.obs, m)
    for (r in idx) {
      cX[r, ] <- corruptor(cX[r,])
    }
    cgm <- geomedian(cX, weight.softmax(cX))
    cwm <- wmean(cX, weight.softmax(cX))

    wmd[k, i] <- euclidean(wm, cwm)
    gmd[k, i] <- euclidean(gm, cgm)
  }
}

wmd <- apply(wmd, 2, function(x) mean(x, na.rm=TRUE))
gmd <- apply(gmd, 2, function(x) mean(x, na.rm=TRUE))

lines(pc, gmd, lty=2, col=2)
lines(pc, wmd, lty=1, col=4)

legend("topleft", legend=c("Weighted Geometric Median (Our approach)",
                           "Weighted Mean"),
      cex=0.8, col=c(2,4), lty=c(2,1), bty = "n")

par(opar)
}

```

SIMULATION STUDY

Synthetic spectra. We first consider synthetic observations whereby $\mu = (1/2, 1/2, 1/2, 1/2, 1/2, 1/2)'$ and the covariance matrix has an "AR(1) structure" given by $\Sigma = \sigma(\rho^{|i-j|})$ for $1 \leq i, j \leq p$ where $\rho > 0$ and $\sigma > 0$ are constants.

```

AR1 <- function(rho, p) {
  Tn <- matrix(0, p, p)
  for (i in 1:p) {
    for (j in 1:p) {
      Tn[i,j] <- rho^abs(i-j)
    }
  }
  return(Tn)
}

```

We set:

```

mu <- rep(0.5, p)
rho <- 0.3
sigma <- 0.01
Sigma <- sigma * AR1(rho, p)

```

Let n be the number of spectra observations.

```
n <- 1000
```

We plot our synthetic observations and the weighted geometric median using our weight model. We see that *without* corruption that the weighted geometric median (gm) and the weighted mean (wm) give similar results.

```
par(mgp = c(3, 0.2, 0), tck=-0.01) -> opar
X <- rtmvnorm(n, mu, Sigma, lb=rep(0, p), ub=rep(1, p))
gm <- geomedian(X, weight.softmax(X))
wm <- wmean(X, weight.softmax(X))
parplot(X, col=alpha("gray", 0.2))
lines(1:p, gm, col=2, lty=2)
lines(1:p, wm, col=4, lty=1)
legend("bottomleft", legend=c("Weighted Geometric Median",
                              "Weighted Mean"),
       cex=0.8, col=c(2,4), lty=c(2,1), bty = "n")
```

```
par(opar)
```

We now define a corruptor function that sets spectra to (1, 1, 1, 1, 1).

```
corruptor <- function(x) {1}
```

As a test, we now corrupt 30% of these observations by setting them to the vector (1, 1, 1, 1, 1). We recall that for surface reflectance, this is the largest possible corruption that can occur. We see that the weighted mean is quite affected by this corruption but our weighted geometric median is not.

```
show.corrupt <- function(corruptor, mu, Sigma, pc=0.30, weight.fn=weight.softmax) {
  par(mgp = c(3, 0.2, 0), tck=-0.01) -> opar
  X <- rtmvnorm(n, mu, Sigma, lb=rep(0, p), ub=rep(1, p))
  gm <- geomedian(X, weight.fn(X))
  wm <- wmean(X, weight.fn(X))

  m <- pc * n
  cX <- matrix(X, nrow=nrow(X))
  idx <- sample(1:n, m)
  for (r in idx) {
    cX[r, ] <- corruptor(cX[r,])
  }

  cgm <- geomedian(cX, weight.fn(cX))
```

```

cwm <- wmean(cX, weight.fn(cX))

parplot(cX[-idx,], col=alpha("gray", 0.2))
parplot(cX[idx,], col=alpha("gray", 0.2), add=TRUE)
lines(1:p, gm, col=1, lty=2)
lines(1:p, cgm, col=2, lty=2)
lines(1:p, cwm, col=4, lty=1)
legend("bottomleft", legend=c("Weighted Geometric Median (Our approach)",
                              "Weighted Mean",
                              "Uncorrupted"),
      cex=0.8, col=c(2,4,1), lty=c(2,1,2), bty = "n")

legend("topright", legend=c(paste0(pc*100, "% Corruption")), bty = "n", cex=0.8)
par(opar)

return(list(gm=gm, wm=wm, cgm=cgm, cwm=cwm))
}

show.corrupt(corruptor, mu, Sigma) -> result

```

We see that the cosine distance between the corrupted cgm and uncorrupted WGM gm is very small:

```
cosine(result$gm, result$cgm)
```

```
## [1] 0.000012638
```

And the Euclidean distance is only slightly affected given such a large amount of corruption:

```
euclidean(result$gm, result$cgm)
```

```
## [1] 0.119
```

On the other hand, the weighted mean is strongly affected by corruption:

```
euclidean(result$wm, result$cwm)
```

```
## [1] 0.35253
```

By looking at the ratio of the two, we see that our weighted geometric median approach is significantly more robust than the weighted mean.

```
euclidean(result$gm, result$cgm)/euclidean(result$wm, result$cwm)
```

```
## [1] 0.33756
```

In this situation, we see that our approach is robust: the cosine distance is very small and the Euclidean distance is clearly within the natural variation of the observations. This should be compared to the weighted mean where the estimated value is well outside the range of variation of the observations.

We note that the above is only a single instance to see the effect of the corruptor on the observations and we now perform a full simulation study to understand the robustness of our approach to the amount of corruption of the observations compared to using a weighted mean approach.

```
sim.study(corruptor, mu, Sigma)
```

FIGURE 1. Corrupting to 1 case: a simulation study show the mean euclidean distance between uncorrupted statistic and corrupted statistic as the percentage level of corruption increases. We see that our approach performs significantly better.

We see that our weighted geometric median (WGM) approach is significantly more robust to this type of corruption compared to the weighted mean approach. As expected, the WGM becomes affected once we approach the breakdown point of the geometric median (50%).

Parallel shift. We now consider parallel shifts of the spectra by choosing our mean to be low and then performing random upwards shifts.

```
mu <- rep(0.4, p)
rho <- 0.3
sigma <- 0.001
Sigma <- sigma * AR1(rho, p)
```

We define a corruptor that takes a spectra, shifts it upwards by 0.2, and clips it to 1 if necessary.

```
corruptor <- function(x) clip(x + 0.2, 0, 1)
```

We show the effect on spectra.

```
show.corrupt(corruptor, mu, Sigma) -> result
```

And now we perform the simulation study.

```
sim.study(corruptor, mu, Sigma)
```

FIGURE 2. Corrupting by parallel shift case: a simulation study show the mean euclidean distance between uncorrupted statistic and corrupted statistic as the percentage level of corruption increases. We see that our approach performs significantly better.

Random spectra. In this case, we set:

```
mu <- rep(0.5, p)
rho <- 0.5
sigma <- 0.001
Sigma <- sigma * AR1(rho, p)
```

We now perform perturb the observations by a truncated normal distributed samples where each band is independent.

```
corruptor <- function(x) rtmvnorm(1, mu, 0.04*diag(1, p, p), lb=rep(0, p), ub=rep(1, p))
```

We show the effect on spectra.

```
show.corrupt(corruptor, mu, Sigma) -> result
```

Finally, we produce the simulation study.

```
sim.study(corruptor, mu, Sigma)
```

FIGURE 3. Corrupting by truncated normals with independent coordinates: a simulation study show the mean euclidean distance between uncorrupted statistic and corrupted statistic as the percentage level of corruption increases. We see that our approach performs significantly better.

References.

- Zhu, Z., & Woodcock, C. E. (2014). Automated cloud, cloud shadow, and snow detection in multitemporal Landsat data: An algorithm designed specifically for monitoring land cover change. *Remote Sensing of Environment*, 152, 217-234.